# Vertical stratification-driven nutrient ratios regulate phytoplankton community structure in the oligotrophic western Pacific Ocean

Zhuo Chen[1,3], Jun Sun[2,3*], Ting Gu[3], Guicheng Zhang[3], Yuqiu Wei[4]

[1] College of Biotechnology, Tianjin University of Science and Technology, Tianjin 300457, China;

[2] College of Marine Science and Technology, China University of Geosciences (Wuhan), Wuhan, Hubei 430074, China;

[3] Research Centre for Indian Ocean Ecosystem, Tianjin University of Science and Technology, Tianjin 300457, China;

[4] Yellow Sea Fisheries Research Institute, Chinese Academy of Fishery Sciences, Qingdao, 266071, China

[*]Correspondence: phytoplankton@163.com

Abstract: Vertical stratification determines the variability of temperature and nutrient distribution in upper seawater, thereby affecting ocean primary production. Nutrients in the oligotrophic region vary in time and space, and thus, phytoplankton vary in their vertical distribution. However, differences in the vertical distribution of phytoplankton have not been studied systematically. This study investigated the spatial distribution pattern and diversity of phytoplankton communities in the western Pacific Ocean (WPO) in the autumn of 2016, 2017, and 2018 and the local hydrological and nutritional status. The Utermöhl method was used to analyze the ecological characteristics of phytoplankton in the surveyed sea area. In the three cruises investigated, we show universal relationships between phytoplankton and (1) vertical stratification, (2) N:P ratio, and (3) temperature and salinity. The potential influencing factors of physical and chemical parameters on phytoplankton abundance were analyzed using a structural equation model (SEM), which determined that the vertical stratification index was the most important influencing factor affecting phytoplankton abundance and indirectly affecting phytoplankton abundance by dissolved inorganic nitrogen (DIN) and dissolved inorganic phosphorus (DIP). Vertical stratification determines the vertical distribution of the phytoplankton community structure in the WPO. The areas with strong vertical stratification (Groups A and B) are more conducive to the growth of cyanobacteria, and the areas with weak vertical stratification (Groups C and D) are more conducive to the bloom of diatoms and dinoflagellates.

Keywords: Vertical stratification; phytoplankton community; western Pacific Ocean; N:P ratio

## 1. Introduction

Phytoplankton account for more than half of global marine primary production and can actively maintain the stability of the entire ecosystem (Sun, 2011). Phytoplankton form the most important level of in the marine food chain (Qian et al., 2005), and changes in their community structure could alter the entire food web. They are widely distributed in various aquatic ecosystems and are marginally affected by fishing activities. Therefore, they are good indicators of marine environment health and climate change (Tang et al., 2017), and studying their community structure is crucial to marine ecology. Phytoplankton growth is closely related to nutrient concentration, and in the stable upper water, the thermocline prevents the upward replenishment of nutrients. Therefore, vertical stratification has a substantial effect on phytoplankton.

As the world's largest and deepest ocean, the Pacific Ocean covers a vast area and has a

complex geographic topography, with the deepest trenches on Earth and the highest absolute peaks (Hu et al., 2016). The study area is located in the western Pacific Ocean (WPO) because the equatorial current flows from east to west. Furthermore, warm seawater in the surface layer flows with the current to the WPO, and in the equatorial region, strong solar irradiation heats the seawater year-round. Under the influence of the dual factors, the average temperature of the ocean surface in the WPO region is higher than 28 °C throughout the year. Through heated seawater, radiant heat, and evaporative heat generated by heated seawater, radiative heat, and latent heat, the WPO is higher than the equatorial eastern Pacific by 3–6 °C, and it has a profound impact on global climate change, especially in China and Southeast Asia (Gordon et al., 1996; Hu et al., 2012; Hu et al., 2015). In addition, the surface primary productivity is low, which is typical of oceans with high temperatures, low salinity, and poor nutrition (Messié et al., 2006). The subtropical Pacific is stratified vertically due to typhoons, upwellings, and various physical mixing processes (Emery et al., 1982). It also causes the sea area to have a 100–150 m thermocline, and the high-temperature seawater in the surface layer transfers heat to the atmosphere through sea-air interaction, which generates large disturbances to the atmosphere, which is the area with the most tropical storms and typhoons formed worldwide (Yan et al., 1992; Wang et al., 2012). Perennial tropical storms and typhoons cause heavy rainfall in the sea, thereby reducing the nutrient content in the sea surface layer and restricting the upwelling of nutrient-rich seawater in the lower layer through the thick thermocline layer and making it difficult to reach the upper water column, which has the typical characteristics of high temperature, high salinity, low nutrient, and low primary productivity (Messié et al., 2006; Kawahata et al., 2002). Due to the Coriolis effect, after the trade wind current reaches the WPO, it encounters the barrier of continental topography and changes its path again, some of which joins the Equatorial Counter Current. However, most of it flows along the continental margin toward higher latitudes and becomes the boundary between the nearshore and oceanic water systems. The Philippine waters at 14–15°N are divided into the southward-moving Mindanao Current (MC) and the northward-moving Kuroshio Current (KC) (Dong et al., 2012; Zhao, 2015). The Kuroshio Current brings high-temperature and high-salinity seawater from the Pacific Ocean to a wide range of offshore sea areas, which greatly impacts the ocean, meteorology, and hydrology of these sea areas. These impacts could result in the movement of fishing grounds, waxing and waning of sea fog, ice conditions in the Bohai and Yellow seas, and even flooding conditions in eastern China, which are all important and related this phenomena (Christian et al., 2004; Gordon et al., 1996). Phytoplankton brought by the Kuroshio water affect the structure of the biotic community in the nearshore area under the effect of multiple factors. During phytoplankton blooms, the subsurface chlorophyll maximum (SCM) usually occurs near or at the bottom of the light-permeable layer of stable seawater (Yentsch, 1965). SCM distribution is closely related to the depth and intensity of the thermocline, and mixing caused by solar radiation and wind is the driving force for regional consistency and latitudinal differences in the thermocline.

Most ocean waters in the global oceans are oligotrophic. With global warming and increased stratification of seawater, these zones are expected to expand, leading to decreases in marine nutrient fluxes and primary productivity (Capotondi et al., 2012; Falkowski et al., 2007; Gruber, 2011). Eutrophic zones with intermittent or irregular nutrient pulses alter the phytoplankton community structure and are ideal for studying changes in phytoplankton community structure dynamics (Lozier et al., 2011; Siokou-Frangou et al., 2010). Changes in seawater stratification and vertical

mixing may affect phytoplankton species composition, abundance, size structure, spatial distribution, phenology, and productivity (Behrenfeld et al., 2006; Daufresne et al., 2009; Edwards et al., 2004). These, in turn, affect the function and biogeochemistry of marine ecosystems (Beaugrand et al., 2009; Hoegh-Guldberg et al., 2010). Therefore, studying the ecological and physiological mechanisms that control changes in the phytoplankton community structure within vertical gradients is essential to assess the response of marine systems to global climate change (Richardson et al., 2004).

Currently, most studies on phytoplankton communities focus on the horizontal distribution at the regional scale, while the vertical stratification of phytoplankton communities has been less studied, and the factors affecting the vertical stratification of phytoplankton remain unclear. The WPO is a typical oligotrophic zone with severe vertical stratification, and seawater stratification has an important influence on the distribution of phytoplankton; therefore, it is necessary to study the vertical stratification of phytoplankton in this region. We investigated how phytoplankton abundance and community composition are related to vertical stratification along a latitudinal gradient in the WPO during 2016–2018. Comparisons between different geographical regions with different vertical density distributions offer a unique opportunity to study how phytoplankton dynamics change as stratification develops.

## 2. Materials and methods

### 2.1. Study area and sampling

This study relied on the shared voyage of the WPO (0–20 °N, 120–130 °E), commissioned by the National Natural Science Foundation of China. Physical, biological, chemical, and geological surveys were carried out on the RV "*kexue*" from September to November in 2016, 2017, and 2018. The sampling stations used in this study are shown in Figure 1; the sampling layers were 5, 25, 50, 75, 100, 150, and 200 m. Phytoplankton samples from different water layers were placed in 1 L PE bottles, fixed in formaldehyde solution (3%), and stored in dark. Nutrient samples from different layers were placed in PE bottles, frozen, and stored at −20 °C for laboratory nutrient analysis.

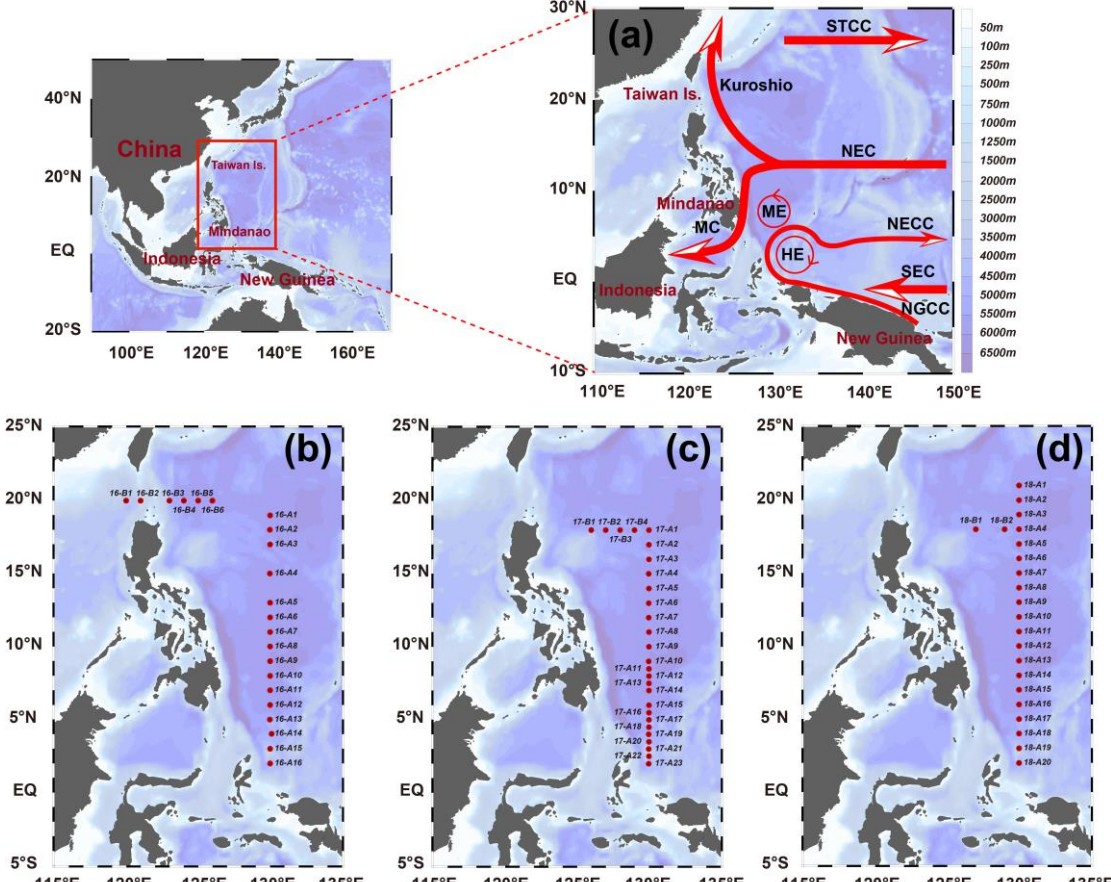

Figure 1. Stations in the western Pacific Ocean (WPO) of three cruises. (a): Current systems of the
WPO; (b), (c), and (d): sampling stations of 2016, 2017 and 2018 cruises, respectively. The station
at 130°E forms the section A, and the station at 20°N forms the section B. Map of the WPO shows
the major geographic names and the surface currents, including the Subtropical Counter Current
(STCC), the North Equatorial Current (NEC), the Northern Equatorial Counter Current (NECC),
the South Equatorial Current (SEC), the New Guinea Coastal Current (NGCC), the Mindanao
Current (MC), the Mindanao Eddy (ME), the Halmahera Eddy (HE).
2.2. Identification of Phytoplankton
After returning to the laboratory, the Utermöhl method was applied for phytoplankton analysis.
A 1 L subsample was allowed to stand for 48 h; then 800 mL supernatant was removed carefully by
siphoning through a catheter, taking care to prevent the catheter from touching the bottom of the
bottle. Thereafter, the remaining 200 mL liquid was gently mixed and half of which was further
concentrated with a 100 mL sedimentation column (Utermöhl method) for 48 h sedimentation. The
phytoplankton species were identified and enumerated under an inverted microscope (AE2000,
Motic, Xiamen, China) at 400× (or 200×) magnification. Phytoplankton identification was
conducted as described by Jin et al. (1965), Isamu Y (1991), and Sun et al. (2002). The World
Register of Marine Species (http://www.marinespecies.org). Species identification was as close as
possible to the species level.
2.3. Nutrient Analysis
The AA3 (SEAL, German) was used for the analysis and determination nutrient. Soluble

inorganic phosphorus ($PO_4$-P) was determined by the phosphomolybdenum blue method with the limit of detection of 0.02 µmol $L^{-1}$; dissolved silicate ($SiO_3$-Si) was determined by the silicon molybdenum blue method with the limit of detection of 0.02 µmol $L^{-1}$; nitrate ($NO_3$-N) was determined by the cadmium column method with the limit of detection of 0.01 µmol $L^{-1}$; nitrite ($NO_2$-N) was determined by the naphthalene ethylenediamine method with the limit of detection of 0.01 µmol $L^{-1}$ (Dai et al., 2008). Ammonia ($NH_4$-N) was determined by the sodium salicylate method with the limit of detection of 0.03 µmol $L^{-1}$ (Guo et al., 2014; Pai et al., 2001). Nitrogen-to-phosphorous (N:P) ratio was calculated by dividing nitrogen concentration ($NO_3^-$+$NO_2^-$) by phosphate concentration.

2.4. Analysis and methods

A SBE911 CTD sensor and standard Sea-Bird Electronics methods were used to process recorded hydrological parameters. The depth of the mixed layer (ML) is calculated as

$$(S, T)= (S_{ref}, T_{ref}-\Delta T)$$

S and T are the average salinity and temperature, respectively, and Sref and Tref are the temperature and salinity at 5 m, $\Delta T$ is equal to 0.5 °C.

We calculated the vertical stratification index (VSI) to indicate the degree of vertical stratification of the water column:

$$VSI=\Sigma\ [\delta_\theta(m+1)-\delta_\theta(m)]$$

where $\delta_\theta$ is the potential density anomaly, and m is the depth from 5 to 200 m.

The abundance of phytoplankton cells in water column was calculated through the trapezoidal integral method (Zhu et al., 2019):

$$P=\left\{\sum_{i=1}^{n-1}\frac{P_{i+1}+P_i}{2}(D_{i+1}-D_i)\right\}/(D_n-D_1)$$

where P is the average value of phytoplankton abundance in water column, P$i$ is the abundance value of phytoplankton in layer $i$, $i + 1$ is the layer $i + 1$, D$n$ is the maximum sampling depth, D$i$ is the depth of layer $i$, and n is the sampling level.

We clustered all species based on Bray-Curtis similarity distance for three years, and the results showed four distinct regions using the Primer (version 6). Distance-based Redundancy analysis (db-RDA) and Principal Co-ordinates Analysis (PCoA) were performed using the R package vegan (version 2.5-7) (Oksanen et al., 2020) to explain the relationship between the environmental parameters (temperature, salinity, depth, VSI, Dissolved inorganic nitrogen (DIN) and Dissolved inorganic phosphorus (DIP) and Dissolved silicate (DSi)) and phytoplankton community structure. The results were visualized using the R package ggplot2 (version 3.3.2). SEM was used to assess the relative direct and indirect impact of physical and chemical parameters on phytoplankton abundance. The chi-square test ($\chi^2$), comparative fit index (CFI), and goodness fit index (GFI) were used to assess the model fit.

3. Results

3.1 Hydrographic features of the study area during the sampling years

The surface temperature and salinity of the surveyed sea area in 2016, 2017, and 2018 are shown in Figure 2. In general, the temperature increased with decreasing latitude, and the stations

near the equator exhibited the highest temperature; in constrast, the salinity showed an opposite trend as that of temperature, with a high value from 15 °N to 20 °N. The surface temperature (Fig. 2) of the surveyed area in 2016 ranged from 28.58 °C (station 16-B1) to 30.14 °C (station 16-A16), with an average of 29.43 °C. The surface salinity (Fig. 2) of the surveyed area in 2016 ranged from 33.80 (station 16-B2) to 34.65 (station 16-A2), with an average of 34.32. The surface temperature (Fig. 2) of the surveyed area in 2017 ranged from 27.91 °C (station 17-A4) to 30.19 °C (station 17-A20), with an average of 29.26 °C. The surface salinity (Fig. 2) of the surveyed area in 2017 ranged from 33.38 (station 17-A16) to 34.64 (station 17-B4), with an average of 33.94. The surface temperature (Fig. 2) of the surveyed sea area in 2018 ranged from 26.33 °C (station 18-B1) to 29.79 °C (station 18-A17), with an average of 28.83 °C. The surface salinity (Fig. 2) of the surveyed sea area in 2018 ranged from 33.77 (station 18-A14) to 34.64 (station 18-B1), with an average of 34.21.

The profile distribution of temperature and salinity based on the cross-sectional data of different water layers at each station obtained from the survey is shown in Figure 2. The temperature of the shallow water column (0–100 m) is higher than that of the deep-water column (100–200 m). The salinity values of the deep-water bodies (100–200 m) were higher than those of the shallow water bodies (0–100 m). The values of temperature and salinity in 2016, 2017, and 2018 did not change significantly. The temperature of the section in 2016 ranged from 12.16 °C (200 m at station 16-A11) to 30.14 °C (5 m at station 16-A16), with an average of 25.74 °C. The salinity of the section in 2016 ranged from 33.80 (5 m at station 16-B2) to 35.39 (150 m at station 16-A16), with an average of 34.61 °C. The temperature of the section in 2017 ranged from 11.16 °C (200 m at station 17-A13) to 30.19 °C (5 m at station 17-A20), with an average of 25.18 °C. The salinity of the section in 2017 ranged from 33.38 (5 m at station 17-A16) to 35.24 (150 m at station 17-A23), with an average of 34.46. The temperature of the section in 2018 ranged from 9.65 °C (200 m at station 18-A14) to 29.79 °C (5 m at station 18-A17), with an average of 24.22 °C. The salinity of the section in 2018 ranged from 33.77 °C (5 m at station 18-A14) to 35.39 °C (150 m at station 18-A17), with an average of 34.57.

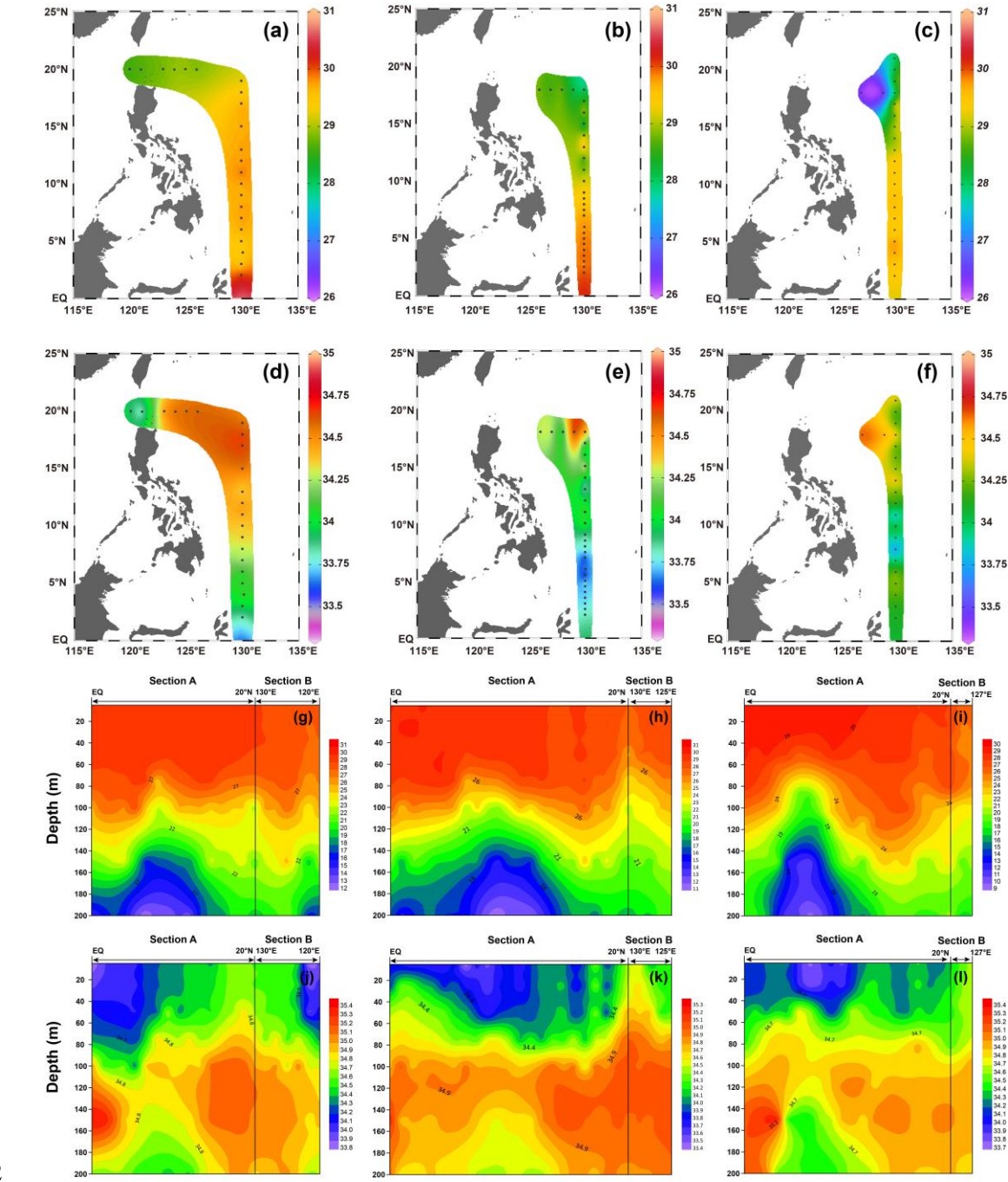

Figure 2. The temperature and salinity distribution in the WPO from three cruises. (a–c) surface temperature in 2016, 2017, 2018 respectively, (d–f) surface salinity in 2016, 2017, 2018 respectively, (g–i) vertical distribution of temperature in 2016, 2017, 2018 respectively, (j–l) vertical distribution of salinity in 2016, 2017, 2018 respectively.

The distribution of the VSI in latitude for the three cruises is shown in Figure 3. Overall, the VSI showed the same distribution pattern in the three cruises, with the highest value occurring at 7–8 °N and a decreasing trend with increasing latitude. In the 2016 cruise (Figure 3-a), the minimum value of VSI (2.54) appeared in the station at 20 °N (station 16-B4), and the maximum value (4.94) appeared in the station at 7 °N (station 16-A11), with an average of 3.90 ± 0.76. In the 2017 cruise (Figure 3-b), a minimum value of VSI (2.85) appeared in the station at 18 °N (station 17-B4), and

the maximum value (5.54) appeared in the station at 7 °N. The maximum value (5.54) occurred in the station at 7 °N (station 17-A14) with an average of $4.30 \pm 0.82$. In the 2018 cruise (Figure 3-c), the minimum value of VSI (2.50) occurred in the station at 18 °N (station 18-B1), and the maximum value (5.48) occurred in the station at 8 °N (station 18-A14) with an average of $4.01 \pm 0.95$. Interestingly, the variation in VSI varied significantly across latitudinal regions; the VSI was high from the equator to 10 °N, while it was low at 10–20 °N.

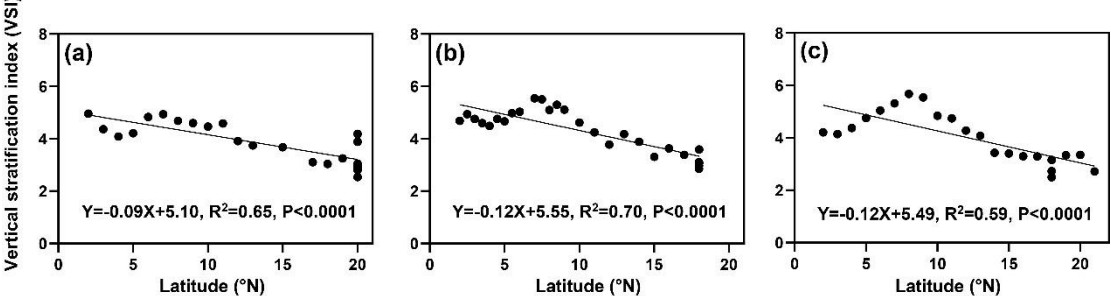

Figure 3. Linear fits of the vertical stratification index with latitude (a) in 2016, (b) in 2017, (c) in 2018. The black dots are the VSI of each station.

3.2. Interannual variability of phytoplankton communities

Figures 4a, b, and c show the horizontal distribution of surface phytoplankton abundance from 2016 to 2018. The interannual variation in phytoplankton was relatively stable, and the sampling area and sampling time from 2016 to 2018 were generally consistent. Most phytoplankton species showed a relatively uniform distribution. Phytoplankton distribution showed a trend of decreasing abundance from the equator to the north, extending in latitude, especially between the equator to 10 °N, where high values of abundance were concentrated. The abnormally high phytoplankton abundance in this region is associated with the predominance of *Trichodesmium*. However, affected by coastal currents, high abundance patches dominated by diatoms were observed in the Luzon Strait area in southern Taiwan, which were carried to the surface by upwelling currents and accounted for more than 67.76% of the abundance at this station. Relatively high abundances were observed at stations in the Kuroshio extension region, consisting mainly of cosmopolitan and warm water species. Phytoplankton abundance was the lowest in the high latitude region.

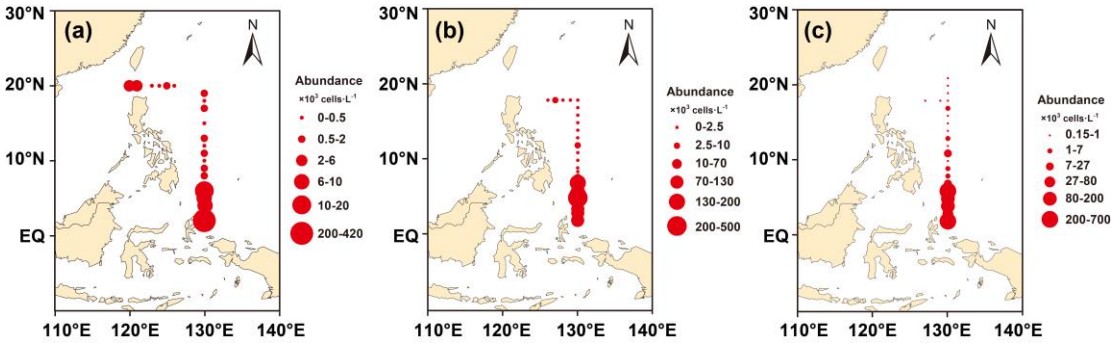

Figure 4. Horizontal distribution of phytoplankton abundance in the WPO. a. 2016, surface layer; b. 2017, surface layer; and c. 2018, surface layer.

3.3. Vertical distribution of phytoplankton abundance

Figure 5 shows the vertical distribution of the phytoplankton. As can be seen from the figure, the overall trend in the WPO was consistent across the three cruises in 2016 (a), 2017 (b), and 2018 (c), with the phytoplankton distribution showing regional variations in latitude and differences in vertical distribution at depth. In terms of latitude, high phytoplankton value areas were concentrated near the equator (0 °E–8 °E), and phytoplankton abundance gradually decreased with increasing latitude. Vertical distribution of phytoplankton indicated that the plankton-abundant areas occurred from 0–50 m, and the phytoplankton abundance gradually decreased with the increase in depth. Vertical distribution of phytoplankton abundance differed significantly across different areas. In the areas near the equator affected by Halmahera Eddy (HE) and Mindanao Eddy (ME), phytoplankton abundance was mainly concentrated in the upper water column (0–50 m) and consisted mainly of cyanobacteria. In the northern area affected by Kuroshio (KC), the phytoplankton abundance was lower than that in the equatorial stations, while the phytoplankton species composition was mostly dominated by diatoms and dinoflagellates.

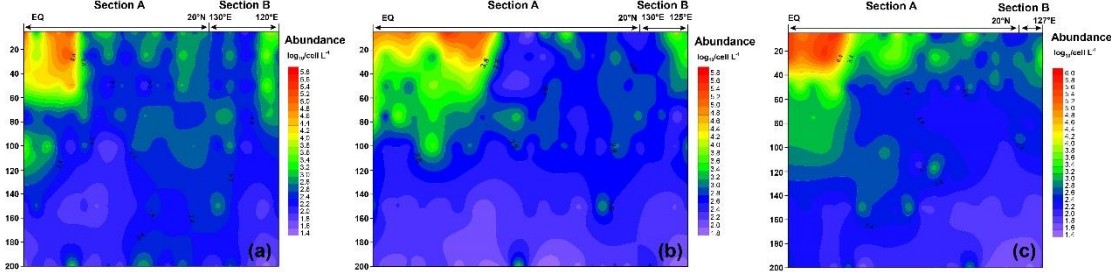

Figure 5. Vertical distribution of phytoplankton abundance (Log10 cells L$^{-1}$) in the WPO in 2016 (a); 2017 (b) and 2018 (c).

3.4. Phytoplankton community structure

Since there was little difference in interannual changes between species, we clustered all species based on Bray-Curtis similarity distance for stations, and the results showed four distinct regions (Figure 6). Cluster analysis divided the phytoplankton communities at the sampling sites for three years into four groups. Cyanobacteria (>90%) were the dominant species in groups A and B. The ratio of diatoms to dinoflagellates in Group A (4.8) was higher than that in Group B (1.4). Cyanobacteria were the dominant (66%) phytoplankton at the stations of Group C, while diatoms (18%) and dinoflagellates (14%) constituted 32% of the population in this group. Diatoms (43%) and dinoflagellates (49%) dominated the stations in Group D, accounting for approximately 92% of the total phytoplankton. The proportion of Chrysophyceae was low in all four groups (Table 1). The dendrogram showed that these populations were grouped into four groups, which were essentially identical to those determined by PCoA analysis (Figure 7). The horizontal and vertical axes explain 51.87% and 21.41% of the phytoplankton community structure, respectively.

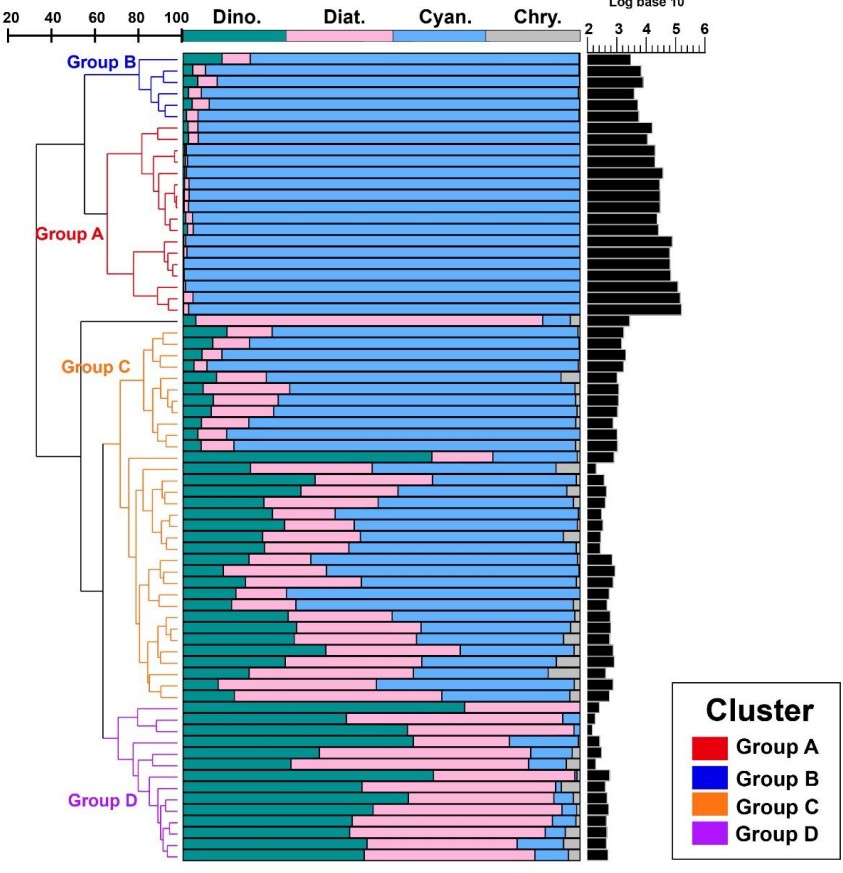

Figure 6. Bray-Curtis similarity-based dendrogram showing averaged phytoplankton community

composition and abundance for each station across the 3 cruises. For each station, community

composition is indicated with bar plots, and phytoplankton abundance is represented with black bars.

Table 1. The percentages (%) (average ± standard deviations) of diatoms, dinoflagellates,

cyanobacteria and Chrysophyceae in the four groups.

| Species | Group A | Group B | Group C | Group D |
|---|---|---|---|---|
| Diatoms | 1.09±0.79 | 4.25±1.57 | 21.83±11.45 | 43.71±10.12 |
| Dinoflagellates | 0.44±0.42 | 3.41±3.30 | 17.26±12.45 | 48.38±11.61 |
| Cyanobacteria | 98.45±1.10 | 92.08±4.79 | 59.05±20.38 | 6.06±4.93 |
| Chrysophyceae | 0.02±0.01 | 0.26±0.10 | 1.86±1.99 | 1.85±1.66 |

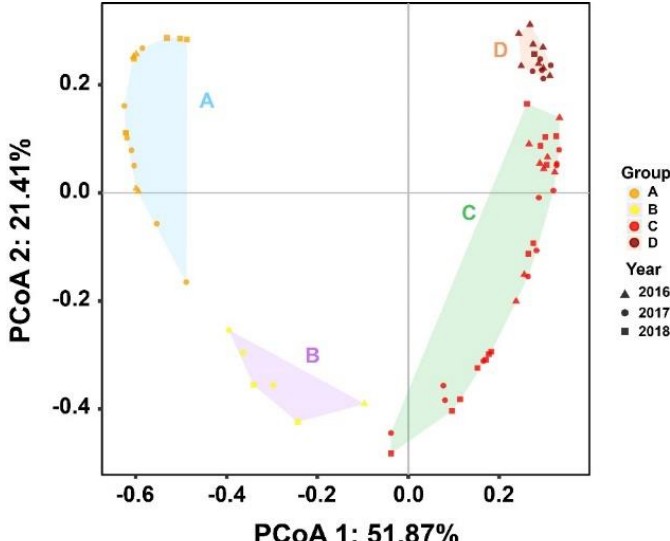

Figure 7. Principal Coordinates Analysis for groups. Triangles, circles, and squares represent 2016, 2017, and 2018 stations, respectively. P < 0.05. Different colors represent different groups. Percentages of total variance are explained by coordinates 1 and 2, accounting for 51.87% and 21.41%, respectively.

3.5. Relationships between phytoplankton and environmental factors

The relationship between phytoplankton and environmental factors was analyzed using RDA. We obtained a two-dimensional distribution map of the species, sample distribution, and environmental factors (Figure 8). The results showed that different phytoplankton classes were correlated differently with environmental factors. Cyanobacteria showed negative correlations with temperature and salinity and positive correlations with VSI and nutrient concentration, indicating that waters with high VSI are suitable for the growth of cyanobacteria (mostly *Trichodesmium*). Diatoms and methanogens showed the opposite trend, exhibiting positive correlations with temperature and salinity and negative correlations with VSI and nutrient concentration, indicating that diatoms and dinoflagellates prefer waters with low VSI.

There were four distinct phytoplankton communities in the WPO: Group A was distributed in the equatorial region with clear vertical stratification. This community is characterized by high abundance and is dominated by *Trichodesmium* species such as *T. thiebautii*, *T. hildebrandtii*, and *T. erythraeum*, which are positively correlated with high concentrations of DIN, phosphate, and silicate. Group B was located near 8°N and is mainly influenced by the NECC and mesoscale eddy influence; the phytoplankton community was represented by warm water species, similar to that of Group A. Group C was mainly distributed in the 15 °N region and was strongly influenced by the NEC. Group D was mainly distributed in the 20°N region, where it was directly influenced by the Kuroshio Current; the phytoplankton community was positively correlated with temperature and salinity.

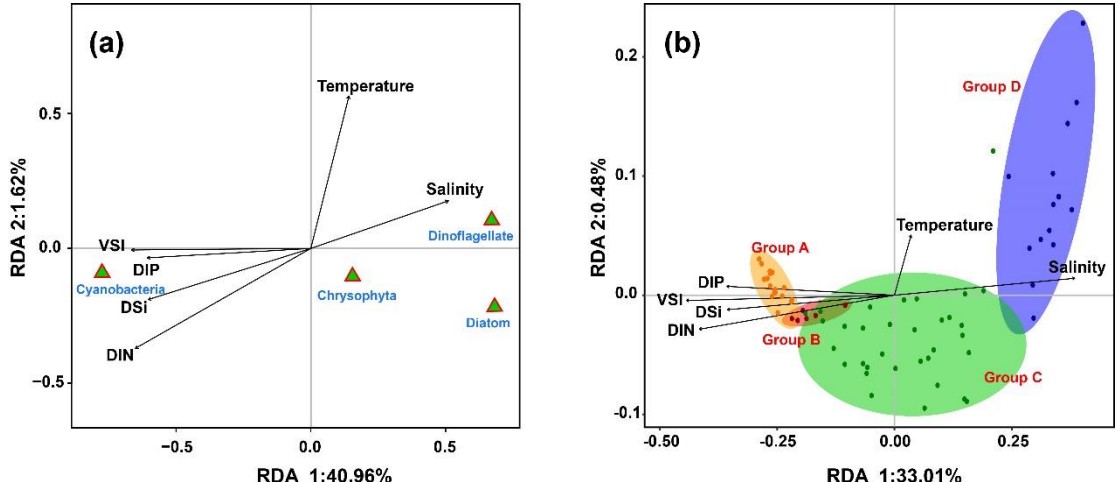

Figure 8. Redundancy analysis of the (a) phytoplankton and environmental parameters, (b) groups and environmental parameters in the WPO. Colored dots represent sampling sites, triangles represent phytoplankton species, and arrows represent environmental factors.

### 3.4 Temperature, salinity, and vertical stratification index

The temperature, salinity, and VSI of the four groups are shown in Figure 9. The temperature and salinity (T-S) box diagram of the sample from 5 m above depicts the four main water masses in the WPO. Groups A (average 29.8 ℃) and B (average 29.6 ℃) had high temperatures, but the salinities of Groups A (average 33.9 °C) and B (average 33.8 °C) was low. The temperature of Groups C (average 28.9 ℃) and D (average 28.9 ℃) was low, but the salinity of Groups C (average 34.2) and D (average 34.4) was high (Fig.9-a). The strong spatial variability of T-S was evident from the characteristics of salinity and temperature. We also calculated the vertical stratification index of the four groups (Fig.9-b). Compared with Groups C (average 3.86) and D (average 3.54), the number of VSIs in Groups A (average 4.69) and B (average 4.86) was markedly higher, and Group A had the highest VSI. There were obvious differences between the four groups; that is, the stratification of the first two groups was more pronounced (Table 2).

The vertical stratification index was linearly fitted to temperature (Fig.9-a) and salinity (Fig.9-b). The fitting results show that the temperature is positively correlated with the vertical stratification index. The VSI of all groups was negatively correlated with salinity. It can be noted that the changes in temperature and salinity were more pronounced in the vertical direction. In Groups A and B with a high stratification index, the changes in temperature and salinity within the group were small. However, the temperature and salinity changed significantly within Groups C and D, with a small stratification index.

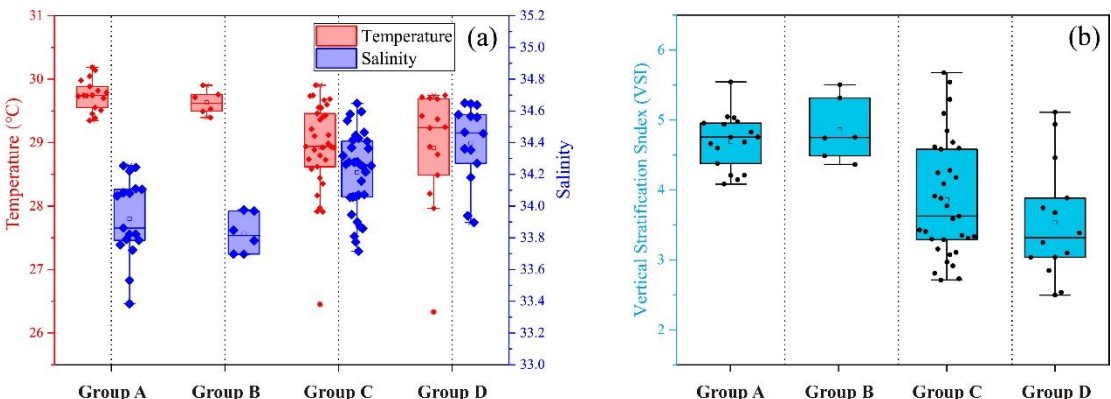

Figure 9 Surface temperature and salinity (a), and vertical stratification index (b) of the four groups.

Table 2. Average (±standard deviations) values for nutrients (μmol L$^{-1}$), temperature (°C), salinity for each phytoplankton community group were identified by the cluster analysis in the WPO.

|  | Group A | Group B | Group C | Group D |
|---|---|---|---|---|
| Temperature | 25.30±1.06 | 24.45±1.85 | 24.92±1.32 | 25.41±1.23 |
| Salinity | 34.45±0.14 | 34.40±0.07 | 34.56±0.16 | 34.68±0.20 |
| DIP | 0.28±0.07 | 0.18±0.13 | 0.16±0.13 | 0.13±0.10 |
| DIN | 4.49±1.76 | 5.43±2.71 | 2.62±1.89 | 1.80±1.08 |
| DSi | 2.93±1.05 | 4.13±2.15 | 1.90±1.47 | 1.44±0.95 |
| VSI | 4.69±0.39 | 4.86±0.45 | 3.86±0.84 | 3.54±0.82 |

3.6. Direct vs. indirect effects of environmental parameter on phytoplankton abundance

The causal relationships between measured phytoplankton abundance and relevant physical and chemical parameters were examined using SEM, using interactions between temperature, salinity, VSI, DIN, and DIP (Fig.10), as theoretical and experimental data indicated the importance of these variables. The model results showed that temperature, DIP, and DIN had a direct effect on phytoplankton abundance, with temperature having the largest direct effect on phytoplankton abundance (0.38), followed by DIN (0.28) and DIP (0.24). Temperature, salinity, and VSI had indirect effects on phytoplankton abundance, with temperature and salinity having negative indirect effects on phytoplankton abundance (-0.17 and -0.30) and VSI having positive indirect effects (0.31) (Figure 10). From the results of the total effect, only salinity had a negative effect on phytoplankton abundance (-0.30), while both temperature and VSI had positive effects on phytoplankton abundance (0.20 and 0.312), with VSI having the largest total effect. Although the direct effect of temperature on phytoplankton abundance was significant, it was partially offset by the indirect negative effect, while VSI had no direct effect on phytoplankton abundance, but its larger indirect effect resulted in the largest total effect. Both DIN and DIP had positive effects on phytoplankton abundance, but the effect of DIN was greater. Since the vertical distribution of DIN and DIP exhibited stronger variability, more specific analyses of DIN and DIP will be conducted later.

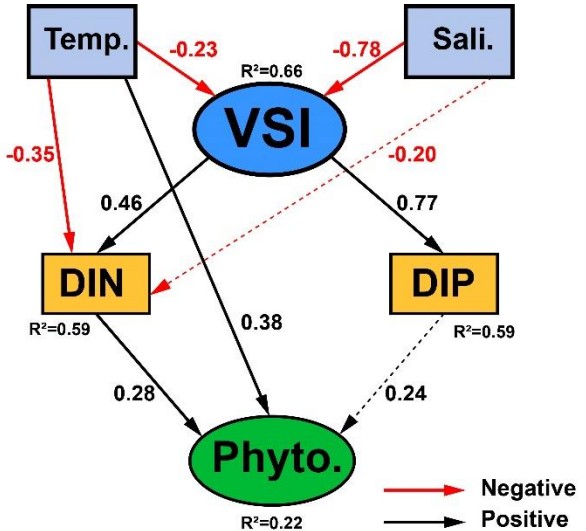

**Chi-square=8.385, p=0.211, CFI=0.989, GFI=0.963**

Figure 10. Structural Equation Model (SEM) analysis examining the effects of temperature, salinity, VSI, DIN and DIP on phytoplankton abundance. Solid black and red lines indicate significant positive and negative effects at p < 0.05, black and red dashed lines indicated insignificant effects. $R^2$ values associated with response variables indicate the proportion of variation explained by relationships with other variables. Values associated with arrows represent standardized path coefficients.

We analyzed the N:P ratio of the surface layer, SCM, and 200 m. The N:P ratio in the surface layer (N:P>16:1) indicates phosphorus limitation, which is consistent with the SEM analysis (Fig.11). The trophic structure of the SCM layer changed, N:P <16:1 indicated nitrogen limitation, and the depth continued to increase to the bottom of the euphotic layer and stabilized around N:P =16:1, indicating that at the bottom of the euphotic layer, as phytoplankton abundance decreased and interspecific competition decreased, the trophic ratio approached the Redfield ratio.

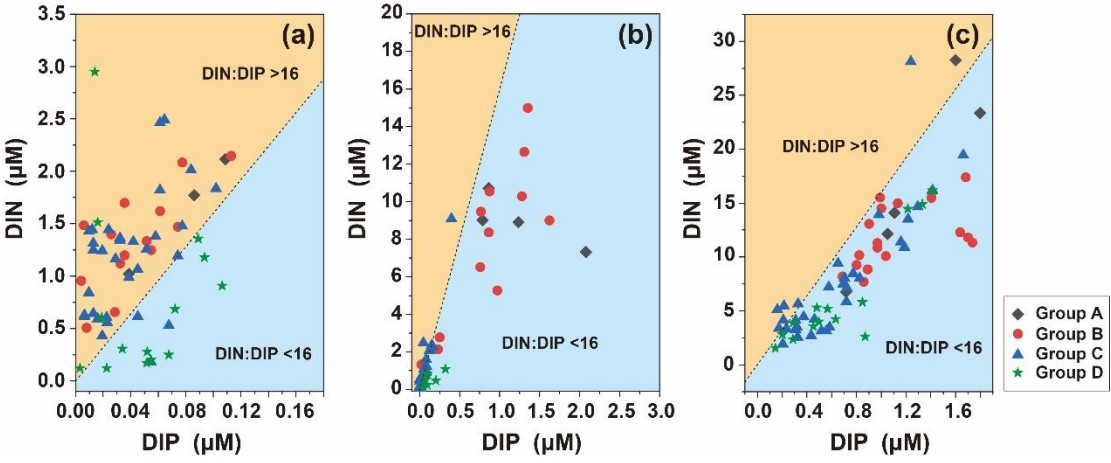

Figure 11. Distribution of phytoplankton community in DIN and DIP. (a): 5 m, (b): SCM, (b): 200 m. The dashed line indicates the Redfield ratio N:P = 16:1.

4. Discussion

4.1. Comparison with historical data

Kuroshio and WPWP are key areas of the WPO sea-air interaction (Zhang, 1999). Previous

surveys have provided less knowledge of the phytoplankton community structure in this study area

(Table 3). Previously, samples were collected by net, and net-collected samples reduced

phytoplankton abundance in small volumes, thereby underestimating the phytoplankton abundance

in the ocean under investigation. In the present study, phytoplankton samples were collected from

water samples, which better reflected the phytoplankton community structure and abundance. Sun

et al. (1997) and Liu et al. (1997) further investigated the species composition and abundance

distribution of phytoplankton diatoms and dinoflagellates in the Ryukyu Islands and nearby waters.

Li et al. (2008) conducted a study on phytoplankton in the tropical and subtropical Pacific oceanic

zones with response mechanisms to the limitation of nitrogen and iron. Chen et al. (2018)

investigated the phytoplankton community structure and mesoscale eddies in the western boundary

current. A total of 199 species in 61 genera belonging to four phytoplankton families were identified,

among which the abundance of *Trichodesmium* species was high. Previous studies have mostly

focused on the vertical trawl and the horizontal distribution of phytoplankton throughout the water

column while ignoring the effect of vertical stratification on phytoplankton.

Table 3. Historical data of the phytoplankton community in the WPO.

| Date | Sampling areas | Layer /m | Number of species | Sampling types | References |
|---|---|---|---|---|---|
| 2018.10 | 2°–20°N, 120°–130°E | 0–200 | 305 | Water samples | This study |
| 2017.10 | 2°–18°N, 126°–130°E | 0–200 | 339 | Water samples | This study |
| 2017.08 | 10.3°–10.9°N, 139.8°–140.4°E | 0–200 | 147 | Water samples | Dai et al.,2020 |
| 2017.08 | 21°–42°N, 118°–156°E | 0–200 | 235 | Water and net samples | Lin et al., 2020 |
| 2017.05 | 21°–42°N, 118°–156°E | 0–200 | 248 | Water and net samples | Lin et al., 2020 |
| 2016.09 | 2°–21°N, 127°–130°E | 0–200 | 269 | Water samples | This study |
| 2016.09 | 0°–20°N, 120°–130°E | 0–200 | 243 | Net samples | Chen et al., 2018b |
| 2014.08 | 0°–21.5°N, 121°–135.5°E | 0–300 | 199 | Net samples | Chen et al., 2018a |
| 1997.07 | 23°30′–29°30′N, 122°30′–130°30′E | 0–200 | 227 | Net samples | Sun et al., 2000 |
| 1997.07 | 23°30′–29°30′N, 122°30′–130°30′E | 0–200 | 251 | Net samples | Liu et al., 2000 |

4.2. Relationship between N:P ratio and vertical distribution of phytoplankton

Research on the factors that control the structure of the phytoplankton community has been

carried out for decades, but the hypothesis of nutrient concentration limits and ratios has not been

fully explained in terms of affecting the structure of the phytoplankton community (Gao et al., 2019).

As diatoms and dinoflagellates show great differences in cell morphology, structure, and nutrition

mode, they differ greatly in their acquisition of nutrient strategies. Several studies have revealed

that dinoflagellates have the ability of mixotrophy, and the mixotrophic modes of dinoflagellates

include direct engulfment of prey, peduncle feeding, and pallium feeding, and phosphorus limitation

is a common factor stimulating dinoflagellates to ingest particulate nutrients (Huang et al., 2005;

Smayda, 1997; Stoecker, 1999). The variation in phytoplankton community structure was always

correlated with fluctuations in physicochemical environmental parameters.

In the four groups we studied, surface seawater N:P>16:1 indicated that phosphorus in surface

seawater was limited, but *Trichodesmium* relied on its own nitrogen fixation function and was highly abundant in oligotrophic waters (Fig.11). The relationship between *Trichodesmium* and nitrogen fixation has been demonstrated several years ago (Grosskopf et al., 2012; Luo et al., 2012; Zehr, 2011). The presence of slight nitrogen limitation in surface seawater in Group D was consistent with the low abundance of *Trichodesmium*, which was consistent with studies on the abundance of *Trichodesmium* in the region (Chen et al., 2019; Sohm et al., 2011). In the WPO, the most oligotrophic ocean around the world (Hansell et al., 2000), nutrients have become an important factor that determines the distribution of phytoplankton. Under nutrition-limited conditions, diatoms and dinoflagellates were more susceptible, especially under phosphorus limitation (Egge, 1998), which corresponds to the high abundance of Group D diatoms and dinoflagellates. In the present study, the vertical pattern of N:P ratios indicated differences in nutrient composition across the vertical gradient. The N:P ratio of the surface layer (N:P>16:1) indicated phosphorus limitation, the structure of nutrients in the SCM layer changed, and (N:P<16:1) indicated nitrogen limitation: the depth continued to increase to the bottom of the euphotic layer and was stable near (N:P=16:1), indicating that at the bottom of the euphotic layer, with decreasing phytoplankton abundance, interspecific competition reduced and the nutrient ratio approached the Redfield ratio. The differences in nutrients partly affected the vertical distribution patterns of phytoplankton abundance. Diatoms have higher phosphorus requirements than other phytoplankton groups, which may be reflected by the lower N:P ratio in diatoms than in other groups (Hillebrand et al., 2013).

4.3. Vertical stratification determined the vertical distribution of phytoplankton

With global climate change, marine oligotrophic regions continue to expand, and seawater stratification is intensified, which is the main problem affecting the marine phytoplankton community structure. The WPO is a typical oligotrophic area with severe stratification. We found that the interannual variation of phytoplankton in stable oligotrophy was not significant, and the intensity of vertical stratification adapted to different environmental changes (nutrients, temperature, and salinity), thus forming four contrasting environments with varying degrees of limiting the community structure of phytoplankton. Comparative analysis of the phytoplankton community composition of the four groups showed that the phytoplankton was mainly strongly affected by the vertical stratification, which corresponds to previous research (Bouman et al., 2011; Hidalgo et al., 2014; Mojica et al., 2015). Vertical stratification limits the replenishment of nutrients in the deep layer and aggravates the formation of the thermocline, which affects the N:P ratio, thereby restricting vertical migration of phytoplankton or affecting the physiology of heat-driven phytoplankton growth and mortality variety (Gupta et al., 2020).

In the present study, *Trichodesmium* was the dominant cyanobacterial species. Marine *Trichodesmium* has been considered the most critical autotrophic nitrogen-fixing cyanobacteria since the 1960s (Dugdale et al., 1961). *Trichodesmium* can be divided into two forms: clusters and free filaments. *Trichodesmium* is suitable for living in waters above 20 °C, and has a special cellular air sac structure that allows it to move vertically within the upper 100 m of the ocean water column (Laroche et al., 2005). In the process of water blooms formed by *Trichodesmium*, a large amount of nitrogen is often fixed in a relatively short period of time. Therefore, the study of the nitrogen fixation rate of *Trichodesmium* is crucial for estimating the rate of nitrogen fixation in the ocean (Karl et al., 2002). Previous studies have not clarified which factors are the main causes of

Trichodesmium growth (possibly temperature, wind, iron, phosphorus, etc.) (Gobler, 2001; Karl et al., 1997; Capone et al., 1997; Chang et al., 2000). Many researchers believe that temperature is the most important factor affecting the growth of *Trichodesmium* (Capone et al., 1999; Adam et al., 2002). However, we believe that there is no single positive correlation between temperature and *Trichodesmium* growth, which is consistent with the study of Chang (2000). In the tropical WPO, where the temperature was not restricted, the abundance of *Trichodesmium* in areas with higher temperatures (Groups A and B) was higher than that at relatively low temperatures (Groups C and D). However, when temperature was no longer the main limiting factor for the growth of *Trichodesmium*, the abundance of *Trichodesmium* in the higher temperature Group A was not higher than the relatively lower temperature Group B as expected, and the high-value area of *Trichodesmium* appeared in Group B. For this reason, we believe that the vertical stratification of Group B was the highest, and the abundance of *Trichodesmium* was also the highest, and the vertical stratification of the four groups was in the following order: Group B > A > C > D (Fig. 9), which was consistent with the high and low abundance of *Trichodesmium* in the four groups. SEM analysis (Fig. 10) revealed that vertical stratification does not directly affect phytoplankton abundance, but indirectly affects phytoplankton growth and abundance by driving the nutrient ratio (N:P). Therefore, we suggest that temperature is not the main factor limiting phytoplankton growth in tropical oligotrophic waters but rather drives nutrient ratios through vertical stratification, which affects phytoplankton growth. This is reflected by the fact that growth of cyanobacteria, diatoms, and dinoflagellates is higher in areas with severe vertical stratification, whereas diatoms and dinoflagellates exhibited superior growth in areas with weak vertical stratification.

Previous models and field experiments have shown that the species composition of phytoplankton communities is significantly affected by vertical turbulent mixing changes (Huisman et al., 2004). A strong coupling exists between the nutrient supply rate and the photosynthetic performance of phytoplankton (Bouman et al., 2006) and phytoplankton biomass and primary production in eutrophic areas are high (Richardson et al., 2019), which directly limits nutrient supply. The vertical stratification index reflects the potential causes of vertical stratification in various physical and chemical processes, such as regulating the utilization of light and nutrients in the ocean, which in turn affects phytoplankton dynamics. The results of the present study showed that from the equator to the north, the VSI decreases as the latitude increases, and the phytoplankton community structure changes from cyanobacteria to diatoms. Phytoplankton abundance was significantly different in the water layer above the SCM. The water layer below the SCM tended to be stable. The surface phytoplankton abundance was usually greater than that of the SCM layer, which was related to the surface layer of *Trichodesmium*. Our results demonstrated that the highly stratified region was more suitable for the growth of *Trichodesmium*, while the region with low vertical stratification seems to be more conducive to the survival of diatoms and dinoflagellates (Fig. 6 and 8). Due to their poor activity and high potential growth rate, diatoms can reproduce rapidly in the circulation and water with high nutrient content (Tilman et al., 1986). The weak vertical stratification of Group C and D regions leads to the homogeneity of temperature, salinity, density, and nutrients in the upper part of 200 m in the vertical direction. The frequency and abundance of dinoflagellates in Groups C and D were higher, which is consistent with the environment where they are more inclined to vertical stratification and weaker (Perez et al., 2006). The vertical distribution of zooplankton has shown that vertical stratification can hinder the migration of small zooplankton populations and indicate

different grazing pressures (Long et al., 2021; Aditee et al., 2005). Further research should consider the difference in predation pressure of different zooplankton predators on the composition of the phytoplankton community in different regions. Phytoplankton stratification may cause thin-layer algal blooms and other phenomena, which was not discussed in this article, and the influence of phytoplankton stratification can be investigated in future studies.

## 5. Conclusions

This study investigated the phytoplankton community structure of the WPO in the autumn of 2016, 2017, and 2018. The WPO is a typical oligotrophic ocean with a weak water exchange capacity owing to the thermocline and severe stratification in the upper seawater layer. The phytoplankton community structure mainly consisted of cyanobacteria, diatoms, and dinoflagellates, while the abundance of Chrysophyceae was low. In terms of spatial distribution, phytoplankton abundance was high from the equatorial region to 10 °N, and decreased with increasing latitude. Phytoplankton showed a high variability in the vertical distribution. The potential influences of physicochemical parameters on phytoplankton abundance were analyzed by SEM to determine nutrient ratios driven by vertical stratification to regulate phytoplankton community structure in a typical oligotrophic. Regions with strong vertical stratification (Groups A and B) were more favorable for cyanobacteria, whereas weak vertical stratification (Groups C and D) was more conducive to diatoms and dinoflagellates.

Funding: This research was financially supported by the National Key Research and Development Project of China (2019YFC1407805), the National Natural Science Foundation of China (41876134, 41676112 and 41276124), the Tianjin 131 Innovation Team Program (20180314), and the Changjiang Scholar Program of Chinese Ministry of Education (T2014253) to Jun Sun.

Acknowledgments: Thank the Natural Science Foundation for its support of the Northwest Pacific voyage for sampling and field experiments. Samples were collected onboard of R/V *Kexue* implementing the open research cruise (voyage number: NORC2016-09, NORC2017-09 and NORC2018-09) supported by NSFC Shiptime Sharing Project. Thank you to all the staff of "*Kexue*" for their help. Thanks for the CTD data provided by Dongliang Yuan Physical Oceanography Research Group, Institute of Oceanography, Chinese Academy of Sciences.

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
