# Peer review of "community structure in the oligotrophic western Pacific Ocean"

_Ocean Science, 2021_

## Author Response (AR1)

Dear Editors and Reviewers:

Thank you for your letter and for the reviewers' comments concerning our manuscript entitled "Vertical stratification-driven nutrient ratios regulate phytoplankton community structure in the oligotrophic western Pacific Ocean" (os-2021-67). Those comments are all valuable and very helpful for revising and improving our paper, as well as the important guiding significance to our researches. We have studied comments carefully and have made correction which we hope meet with approval. The article has been changed considerably, so it is not marked in red in the text. The main corrections in the paper and the responds to the reviewer's comments are as flowing:

Responds to the reviewer's comments:
Reviewer #1:

1. Response to comment: (This manuscript reported the effect of vertical stratification on phytoplankton community structure in the western Pacific Ocean. The topic is interesting, especially under the current situation of global warming. Although the authors accumulated a large amount of data, they fails to provide a convincing story and a novel conclusion. The results and discussions are not closely related. The authors described a lot on the differences among four groups, but they did not discuss much on this point. In turn, some discussed points lacked the supporting results. The whole discussion part lacked depth and logic. It is more like a review rather than a discussion based on the obtained results. There are many errors in the figure legends. There are many errors in the figure legends. Also, there are lots of typos and format and gramma errors in the whole manuscript and need to be carefully checked and corrected.)

Response: Dear Reviewer, thank you very much for your kind letter and encouragement. We also appreciate the time and effort that you have dedicated to providing valuable comments and suggestions, which helped us to improve the quality of our manuscript significantly. We have carefully studied these insightful comments and have made corrections which we hope meet with your approval. Knowing our limited English skills, we sought out a professional English language retouching company to revise the manuscript. If there is anything else we should do, please don't hesitate to let us know. Again, we deeply appreciate your efficient and professional review of our manuscript.

[Figure]

Certificate of Elsevier
Language Editing Services

The following article was edited by Elsevier Language Editing Services:
"Vertical stratification driven nutrient ratios to regulate phytoplankton
community structure in the oligotrophic western Pacific Ocean"

Authored by:
ZhuoChen

Date: 11-Aug-2021
Serial number: LEEX-15215-E4F1D0CAC006

2.   Response to comment: (P2L1-2: Please revise the English in this sentence. "WPO is not only… but also suffer the highest number of ….")

Response: Dear Reviewer, we are very sorry for our poor description. We have revised the sentence: "As the world's largest and deepest ocean, the Pacific Ocean covers a vast area and has a complex geographic topography, with the deepest trenches on Earth and the highest absolute peaks."

3.   Response to comment: (P2L2: "Marine changes"?)

Response: Dear Reviewer, we are very sorry for our poor description. We have revised the sentence: "The study area is located in the western Pacific Ocean (WPO) because the equatorial current flows from east to west. Furthermore, warm seawater in the surface layer flows with the current to the WPO, and in the equatorial region, strong solar irradiation heats the seawater year-round. Under the influence of the dual factors, the average temperature of the ocean surface in the WPO region is higher than 28 °C throughout the year. Through heated seawater, radiant heat, and evaporative heat generated by heated seawater, radiative heat, and latent heat, the WPO is higher than the equatorial eastern Pacific by 3–6 °C, and it has a profound impact on global climate change, especially in China and Southeast Asia."

4.   Response to comment: (P2L9-10: Gramma mistake. Please re-write this sentence)

Response: Dear Reviewer, we are very sorry for our poor description. As your suggestion, we have carefully reworked the grammatical issues of our manuscript in the process of modification. Meanwhile, we have revised the whole manuscript with the assistance from polish company. Again, we deeply appreciate your efficient and professional review of our manuscript. We have revised the sentence: "In addition, the surface primary productivity is low, which is typical of oceans with high temperatures, low salinity, and poor nutrition."

5.    Response to comment: (P2L10-12: "Because of typhoon, upwellings and various kinds of physical mixing processes, vertical stratification of subtropical Pacific seawater" Gramma mistake. The sentence is missing a verb.)

Response: Dear Reviewer, we are very sorry for our poor description. As your suggestion, we have carefully reworked the grammatical issues of our manuscript in the process of modification. As your suggestion, we have revised the sentence: "The subtropical Pacific is stratified vertically due to typhoons, upwellings, and various physical mixing processes."

6.    Response to comment: (P2L14-16: Please re-write the sentence. "wind-induced" is an adj, not a noun.)

Response: Dear Reviewer, we are very sorry for our poor description. As your suggestion, we have carefully reworked the grammatical issues of our manuscript in the process of modification. As your suggestion, we have revised the sentence: "It also causes the sea area to have a 100–150 m thermocline, and the high-temperature seawater in the surface layer transfers heat to the atmosphere through sea-air interaction, which generates large disturbances to the atmosphere, which is the area with the most tropical storms and typhoons formed worldwide."

7.    Response to comment: (P2L16: "Therefore, it is characterized by SCM regions by latitude." How did the authors draw this conclusion?)

Response: Dear Reviewer, thank you very much for your valuable and constructive comments. We are very sorry for your confusion due to our poor description, we have revised this sentence into "SCM distribution is closely related to the depth and intensity

of the thermocline, and mixing caused by solar radiation and wind is the driving force for regional consistency and latitudinal differences in the thermocline."

8. Response to comment: (P2L16-18: Please provide the reference. Moreover, "the TTS.." should be "The TTS..")

Response: Dear Reviewer, thank you very much for your valuable and constructive comments. We have revised the capitalization of the first letter of the sentence. Thank you again for your insightful comments, and we will pay attention to this issue in our future writing.

9. Response to comment: (P2L18-19: What is ternary input? Please explain.)

Response: Dear Reviewer, thank you very much for your careful and professional review of our manuscript, and your valuable comments are very important for us to improve the accuracy and quality of our manuscript. We deleted this sentence due to inaccurate expression.

10. Response to comment: (P2L19-20: "The SCM in the tropical WPO is 80 m (Dandonneau, 1979)." Please provide more evidence. I believe the SCM should be different in different regions in WPO.)

Response: Dear Reviewer, thank you very much for your insightful and constructive comments. Here you have raised a very important point, and we absolutely agree with your point. Thanks to your comments, we have revised the sentence: "During phytoplankton blooms, the subsurface chlorophyll maximum (SCM) usually occurs near or at the bottom of the light-permeable layer of stable seawater."

11. Response to comment: (Method: Section 2.2: Although the authors cited some references, the method should be still described briefly.)

Response: Dear Reviewer, thank you very much for your valuable and constructive comments. The description of the method in the original text is too simplistic, and the method of investigation and experiment is not elaborated. We have supplemented it and read as follows:

2.2. Identification of Phytoplankton

After returning to the laboratory, the Utermöhl method was applied for phytoplankton analysis. A 1 L subsample was allowed to stand for 48 h; then 800 mL supernatant was removed carefully by siphoning through a catheter, taking care to prevent the catheter from touching the bottom of the bottle. Thereafter, the remaining 200 mL liquid was gently mixed and half of which was further concentrated with a 100 mL sedimentation column (Utermöhl method) for 48 h sedimentation. The phytoplankton species were identified and enumerated under an inverted microscope (AE2000, Motic, Xiamen, China) at 400× (or 200×) magnification. Phytoplankton identification was conducted as described by Jin et al. (1965), Isamu Y (1991), and Sun et al. (2002). The World Register of Marine Species (http://www.marinespecies.org). Species identification was as close as possible to the species level.

12. Response to comment: (Result: I suggested the authors adding a paragraph to introduce the hydrographic features of the study area during the sampling years.)

Response: Dear Reviewer, thank you very much for your careful and professional review of our manuscript, and your valuable comments are very important for us to improve the accuracy and quality of our manuscript. We have added a paragraph to introduce the hydrological characteristics of the study area during the sampling years. "3.1 Hydrographic features of the study area during the sampling years."

13. Response to comment: (Fig 1: The figure legend is not correct. There are no red, yellow and green triangles, and black dots in the figure. I guess fig b,c,d are sampling maps in 2016, 2017, and 2018, respectively?)

Response: Dear Reviewer, we are very sorry for your confusion due to our poor description, we have corrected the legend in Figure 1.

Figure 1. Stations in the western Pacific Ocean (WPO) of three cruises. (a): Current systems of the WPO; (b), (c), and (d): sampling stations of 2016, 2017 and 2018 cruises, respectively. The station at 130°E forms the section A, and the station at 20°N forms the section B. Map of the WPO shows the major geographic names and the surface currents, including the Subtropical Counter Current (STCC), the North Equatorial Current (NEC), the Northern Equatorial Counter Current (NECC), the South Equatorial Current (SEC), the New Guinea Coastal Current (NGCC), the Mindanao Current (MC),

the Mindanao Eddy (ME), the Halmahera Eddy (HE).

14. Response to comment: (Fig 2: I suggest the authors adding the taxa information in the Fig. 2. The figure legend is confusing. There were no subregions in the figures, but only three years. Also, a scale bar should be added.)

Response: Dear Reviewer, thank you very much for your valuable and constructive comments. We are very sorry for our poor description, we deleted Figure 2 and replaced it with Figure 3–5.

15. Response to comment: (Fig. 3: How did the author average the phytoplankton community structure? By depth integration?)

Response: Dear Reviewer, we would like to express our great appreciation to your time and effort in reviewing our manuscript, and your insightful and constructive comments helped to improve the accuracy and quality of our manuscript significantly. We have studied these insightful comments and made careful revisions which we hope meet with your approval. We averaged the abundance of the 7-layer samples at each station. We have supplemented the formula used for averaging in the method section 2.4. If there is anything else we should do, please don't hesitate to let us know. Again, we deeply appreciate your efficient and professional review of our manuscript.

$$P=\left\{\sum_{i=1}^{n-1}\frac{P_{i+1}+P_i}{2}(D_{i+1}-D_i)\right\}/(D_n-D_1)$$

where P is the average value of phytoplankton abundance in water column, $P_i$ is the abundance value of phytoplankton in layer $i$, $i + 1$ is the layer $i + 1$, $D_n$ is the maximum sampling depth, $D_i$ is the depth of layer $i$, and n is the sampling level.

16. Response to comment: (Table 1: Please add the standard deviations.)

Response: Dear Reviewer, thank you very much for your professional and careful review of our manuscript and for giving constructive comments. We added the standard deviations to Table 1.

Table 1. The percentages (%) (average ± standard deviations) of diatoms, dinoflagellates, cyanobacteria and chrysophyceae in the four groups respectively.

| Species | Group A | Group B | Group C | Group D |
|---|---|---|---|---|

| | | | | |
|---|---|---|---|---|
| Diatoms | 1.09±0.79 | 4.25±1.57 | 21.83±11.45 | 43.71±10.12 |
| Dinoflagellates | 0.44±0.42 | 3.41±3.30 | 17.26±12.45 | 48.38±11.61 |
| Cyanobacteria | 98.45±1.10 | 92.08±4.79 | 59.05±20.38 | 6.06±4.93 |
| Chrysophyceae | 0.02±0.01 | 0.26±0.10 | 1.86±1.99 | 1.85±1.66 |

17. Response to comment: (P5L19-20: PCoA analysis cannot explain the relationship between the environmental parameters and phytoplankton community structure.)

Response: Dear Reviewer, we are very sorry for your confusion due to our poor description. It is true as you presented that PCoA analysis cannot explain the relationship between environmental parameters and phytoplankton community structure. It can be used to analyze the differences between species groups. The revised sentence is: "The dendrogram showed that these populations were grouped into four groups, which were essentially identical to those determined by PCoA analysis (Figure 7). The horizontal and vertical axes explain 51.87% and 21.41% of the phytoplankton community structure, respectively."

18. Response to comment: (Fig. 5: What are differences among the panels a,b,c,d? Please add the description for each panel.)

Response: Dear Reviewer, we are very sorry for our poor description. The four pictures represent the four Groups, the expression is not clear. We deleted this figure and replaced it with figures 4 and 5.

19. Response to comment: (Fig. 6&7: I think these two figures only included surface temperature and salinity. Please add this information in the figure legend.)

Response: Dear Reviewer, we are very sorry for our poor description. As your suggestion, the figures only included surface temperature and salinity, we add this information in the figure legend.

20. Response to comment: (P8L16-17& Fig.7: This figure did not make much sense, and it was not discussed in the discussion part.)

Response: Dear Reviewer, thank you very much for your valuable and constructive comments, we deleted the figure.

21. Response to comment: (P9L12: statistical analysis is needed to prove the "significant differences" in phytoplankton community structure across groups.)

Response: Dear Reviewer, thank you very much for your valuable and constructive comments. We added an RDA diagram of phytoplankton and the environment, and we re-describe the RDA (Results 3.5). We apologize for any inconvenience caused to your review.

22. Response to comment: (Discussion P12L11-13: "It can be seen that the density of Trichodesmium in Kuroshio region was very high..". I did not see the data of trichodesmium in the Kuroshio region. I suggest the authors providing the detailed phytoplankton community structure.)

Response: Dear Reviewer, thank you very much for your professional and careful review of our manuscript and for giving constructive comments. I apologized for any inconvenience caused to your review here, the area we surveyed is the Kuroshio source area, and we want to express the high abundance of *Trichodesmium* in the surveyed area. We previously focused on the effect of vertical stratification on phytoplankton, and did not have a large description of the horizontal distribution. We supplemented the horizontal distribution in the results section to better describe the phytoplankton community structure in the surveyed sea area (Figure 4 and 5). We have revised the sentence in discussion 4.1: "among which the abundance of *Trichodesmium* species was high."

23. Response to comment: (P13L35-37: The result part did not show the change of VSI and phytoplankton community structure with latitude. If this is an important point of this paper, the relevant result should be added.)

Response: Dear Reviewer, we are very sorry for your confusion due to our poor description. It is true as you presented that we did not calculate VSI and phytoplankton community structure with latitude throughout the paper, but in the discussion, we kept focusing on it, this was indeed an oversight on our part. Thanks to your comments, we

added the change of VSI and phytoplankton community structure with latitude (Result 3.1-3.3 and Figure 3-5).

24. Response to comment: (P13L40-42L: Those results should be presented in figures and described in the result part.)

Response: Dear Reviewer, thank you very much for your professional and careful review of our manuscript and for giving constructive comments. We have added details in section 3.3 and 3.4. And we have revised the sentence into: "Our results demonstrated that the highly stratified region was more suitable for the growth of *Trichodesmium*, while the region with low vertical stratification seems to be more conducive to the survival of diatoms and dinoflagellates (Fig. 6 and 8)."

25. Response to comment: (P13L43-44: Please provide the reference.)

Response: Dear Reviewer, thank you very much for your valuable and constructive comments. We have listed the reference and we have revised the sentence: "Due to their poor activity and high potential growth rate, diatoms can reproduce rapidly in the circulation and water with high nutrient content (Tilman et al., 1986)."
Tilman, D., Kiesling, R., Sterner, R., Kilham, S. S., and Johnson, F. A.: Green, bluegreen and diatom algae: taxonomic differences in competitive ability for phosphorus, silicon and nitrogen, Arch Hydrobiol, 106(4): 473-485, doi: 10.1029/WR022i007p01162, 1986.

Reviewer #2:

Response to comment: (The manuscript was clearly written by a colleague with insufficient command of English. As well as making many parts of the manuscript impossible to understand precisely, it seems that the poor command of English has led the author to make statements that are unnecessary and/or incorrect.
On the other hand, the aim of the investigation seems original and important.
I know that amongst the co-authors, there is talent to do much better than this! I suggest they collectively re-write the manuscript with care, and have the English professionally polished before resubmitting.)

Response: Dear Reviewer, thank you very much for your kind letter and encouragement. We also appreciate the time and effort that you have dedicated to providing valuable comments and suggestions, which helped us to improve the quality of our manuscript significantly. We have carefully studied these insightful comments and have made corrections which we hope meet with your approval. Knowing our limited English skills, we sought out a professional English language retouching company to revise the manuscript. If there is anything else we should do, please don't hesitate to let us know. Again, we deeply appreciate your efficient and professional review of our manuscript.

[Figure]

Special thanks to you for your good comments.

---

## Referee Report (RR1)

Review on

**General aspects**

This paper reports the Utermöhl plankton in the Tropical West Pacific at almost identical stations across latitudes from the equator to 20°N., in the autumn of three successive years, 2016, 2017 and 2018, aboard the flagship of the PLC's ocean research fleet, the R.V. *Keshue*. Identification was reported only according to broad groups. Samples were taken throughout the water column from 5 to 200 m. Over an essentially 2D transect from south to north, with a minor dog leg north of the Philippines.

The taxonomic and physico-chemical data were explored by several statistical tools, the Structural Equation Model (SEM), PCA, RDA, and Bray-Curtiss Analysis. As well as fitting the standard physico-chemical parameters of nitrate, nitrite, phosphate, silicate, T and S, the authors systematically computed a Vertical Stratification Index (VSI), including this parameter in statistical treatments.

The paper presents: T/S data as x-y surface and x-z sections for each of the 3 years (Fig. 2); variation in VSI from south to north over the 3 years (Fig. 3); surface phyto abundance (Fig. 4); x-z distribution of phyto abundance over 3 years (Fig. 5); broad taxonomic relative abundance distribution (dinos, diatoms, cyanos, chrysos) over all stations, pooled for the 3 years (Fig. 6), 2D PCA diagram of the stations (Fig. 7); 2D RDA diagrams (Fig. 8). These analyses are used by the authors to show clearly that the study area divides into 4 groups, A, B, C and D. In Fig 9 whisker boxes are used to show the distribution of T, S and VSI in each group, which is a very nice feature, and very clearly presented. Fig. 10 is used to present results of the SEM, indicating the statistically computed quantified effects (apparent effects?) of T and S on VSI (of course) and of T, S and VSI on DIN, DIP and phytoplankton. This is very original, as far as I am aware. Fig. 11 is used to explore the effects of DIN and DIP (particularly the N:P ratio, on phytoplankton of the four major taxa in the 4 regions at 3 depths (surface, DCM, and 200 m). It clearly shows different effects of N and P on the phytoplankton community structure in the different ecosystems corresponding to these three chosen depths.

There follows a Discussion rich, original and well argued.

However, in contrast to all this quality the introduction is terrible, and totally inappropriately targeted. While the authors have done an excellent job for the Methods, the Results and the Discussion, they need to scrap the Introduction completely and write it again. The same goes for the Abstract. If this is done well, this manuscript would constitute an important and original contribution.

**Specific aspects**

**Title**

OK

**Abstract**

TERRIBLE.

**Introduction**
TERRIBLE. (See above).

**Materials and Methods**
THIS SECTION IS MOSTLY EXCELLENT.

P3L26 "PE" > "polyethylene"

P4L11 Insert reference for the Utermöhl method.
L11-20 In this section state the minimum size of organisms identified and counted

L23 "AA3 (SEAL., German(y)" Give bibliographic reference or web site.

P5L15 "average" of temperature and salinity: give the precise dataset for which the average was computed.
L28 "the three years"

**Results**
THIS SECTION IS MOSTLY EXCELLENT.
P7L9 "the same" > "a similar"

P8L5 Delete "variation in" (repetition)
L12 "variability" > "variation"
L16 "showed a relatively uniform" > "varied little from year to year in their"
L17-18 "extending..." > with a minor abundance peak at about 10°N."
L18-19 "The abnormally..." > "This abundance peak was associated..."
L20 "observed also..."
L21 "in southern Taiwan" > "south of Taiwan"

P9L1 Delete «As can be seen from the figure,"
L3 "regional variations in latitude" > "variations with latitude"
L5-6 Delete ", and..." (Repetition)
L11-13 "...the lower phytoplankton abundance was mostly dominated by..."
L19 "... little interannual difference between species, ..."
L26 "4.8" > "4.8%"; "1.4" > "1.4%"
L29-30 Delete "The horizontal.." It's already marked on the figure 7.

P11 L14 "methanogens" This seems to be a mistake.
L26 Insert "here," before "the phytoplankton"

P12L6 "3.4" > "3.6"
L8 "... of the sample from 5 m above" Seems to be a spurious insertion. Delete.
L12-13 "The strong spatial variability..." > "Fig. 9 shows clear variation in T-S.".
L14-15 number of VSIs" >"values of VSI"
L15 "was > "were"
L16 Delete "There were... groups"
L18 "linearly fitted to temperature" > "related to temperature"
L19 Delete "The fitting results showed that the"
L20-21 Delete "It can be noted that the"
L21 "more" > "most"

P13L7 "3.6" > "3.7"
L7 "parameter" > "parameters"

P14L14 At the end of this light you may like to add, "and growth may have become increasingly limited by light."

**Discussion**
VERY GOOD DISCUSSION.

P15L1 "Kuroshio" > "The Kuroshio"
L1 "WPWP" Add this to Fig. 1.
L1 After "interaction" insert "and climate modulation"
L14 "the vertical trawl" > "vertical hauls"
L24 "acquisition of nutrient strategies" > " nutrient acquisition strategies"

L25-26 "... dinoflagellates use mixotrophy, engulfing prey as well as feeding using peduncles and palia, while phosphorus..."
L28 "was > "is"

P16L2 "Fig. 11" > "Fig. 6"
L3-4 "... has already been demonstrated (Grosskopf et al., 2012; ..."
L4 "The presence of slight" > "The virtual absence"
L9 "were" > "are"
L9 "susceptible" > "affected"
L11 "across" > "along"
L12 "indicated" > "indicates"
L13 "indicated" > "indicates"
L16 "competition is reduced as light limitation kicks in, and the nutrient ratio approaches...
L17 "nutrients partly affected..." > "nutrient ratios thus affect..."
L20 Around here, it might be good to very briefly mention the possibility of limitation by other nutrients such as iron. Also mention, if you like, that some of the phytoplankton sampled may have recently sunk from upper layers, and therefore represent the nutrient rations and T-S of these layers. You, the authors, may have a feeling for this in the present work.
L22-24 Delete "With global... structure". It's too speculative.
L24 Delete "typical"
L24 "severe" > "strong"
L25-28 "... the interannual variation of phytoplankton was not significant. It remained stably oligotrophic, and the vertical stratification structure determined that of environmental resources such as nutrients, thus forming four contrasting environments, each with its characteristic phytoplankton community structure." [I think you can't say that stratification produced the T-S environment. In any case there is no need to say it.]
L31-34 "... from the deep layer below the thermocline, which affects the N:P ratio, and restricts vertical migration as well as physiologically affecting the vertical structure of phytoplankton growth and mortality."
L37 "has been" > "is"
L38 Delete "since the 1960s"
L39 "is suitable for living > "thrives"

P17L1 "*Trichodesmium*" needs italics
L2 "believe" > "have proposed"
L4 "believe" > "suggest"
L5 "also is consistent"
L6 "where the temperature was not restricted," > "where the surface temperatures all exceeded 20°C.,"
L7 "higher than in those with lower temperatures"

L8-21 based on the data you present and other knowledge, the present referee is not entirely convinced by the authors' arguments, but the authors should have the right to interpret their data in this way if they so choose.

L10 "high-value area" > "high abundance"
L20 "severe" > "strong"
L25-27 "A strong coupling exists among the nutrient supply rate, the photosynthetic performance of phytoplankton (Bouan et al., 2006), the phytoplankton biomass and primary production, particularly in eutrophic areas (Richardson et al., 2019)." Delete "which directly limits nutrient supply"
L28 "causes" > "effects"
L28 "in" > "on"
L34 "which" > "and"
L35 "demonstrated" > "demonstrate"
L37 Delete "the survival of"
L38 "poor activity" Do you mean "low mobility"?

L38-39 "in the circulation and water with high nutrient content" > "in mixed water with high nutrient content"
L40 After "C and D regions", insert "(Fig. 9b)"
L40 "the" > "relative"
L41-43 I don't understand what you mean in this sentence.

P18 L4 Delete "which was not discussed in this article"
L5 "further investigated"

**Conclusions**
P19L9 Delete "typical"
L14 "variability" > "variation"
L15 "Structural Equation Model (SEM)"
L17 After "oligotrophic", insert "sea area"

**Acknowledgements**
L26 "We thank"

**References**
I not that given names and family names are inverted in the first reference ,"Mitra, A. and Flynn, K.J."
Please check all references.

**Figures, Tables**
The figures and tables are all very good.
Fig. 7 needs more contrast.

---

## Author Response (AR2)

Dear Editors and Reviewers:

Thank you for your letter and for the reviewers' comments concerning our manuscript entitled "Vertical stratification-driven nutrient ratios regulate phytoplankton community structure in the oligotrophic western Pacific Ocean" (ID: os-2021-67). Those comments are all valuable and very helpful for revising and improving our paper, as well as the important guiding significance to our researches. We have studied comments carefully and have made correction which we hope meet with approval. Revised portion are marked in red in the paper. If there is anything else we should do, please don't hesitate to let us know. Again, we deeply appreciate your efficient and professional review of our manuscript. The main corrections in the paper and the responds to the reviewer's comments are as flowing:

Responds to the reviewer's comments:

Reviewer #1:

**General aspects**

This paper reports the Utermöhl plankton in the Tropical West Pacific at almost identical stations across latitudes from the equator to 20°N., in the autumn of three successive years, 2016, 2017 and 2018, aboard the flagship of the PLC's ocean research fleet, the R.V. Keshue. Identification was reported only according to broad groups. Samples were taken throughout the water column from 5 to 200 m. Over an essentially 2D transect from south to north, with a minor dog leg north of the Philippines.

Response: Dear Reviewer, thank you very much for your kind letter and encouragement. We also appreciate the time and effort that you have dedicated to providing valuable comments and suggestions, which helped us to improve the quality of our manuscript significantly. We have carefully studied these insightful comments and have made corrections which we hope meet with your approval. In this paper, we identify reported only according to broad groups. Because our study found that the correlation of phytoplankton with VSI and physicochemical factors was more intuitively expressed at the phylum level in the study area. We wanted to express this scientific question prominently, so we did not discuss community diversity in detail. All identified species belong to four phyla (Bacillariophyta, Dinophyta, Cyanophyta, and Chrysophyta).

The taxonomic and physico-chemical data were explored by several statistical tools, the Structural Equation Model (SEM), PCA, RDA, and Bray-Curtiss Analysis. As well

as fitting the standard physico-chemical parameters of nitrate, nitrite, phosphate, silicate, T and S, the authors systematically computed a Vertical Stratification Index (VSI), including this parameter in statistical treatments.

Response: Dear Reviewer, we would like to express our great appreciation to your time and effort in reviewing our manuscript, and your insightful and constructive comments helped to improve the accuracy and quality of our manuscript significantly.

The paper presents: T/S data as x-y surface and x-z sections for each of the 3 years (Fig. 2); variation in VSI from south to north over the 3 years (Fig. 3); surface phyto abundance (Fig. 4); x-z distribution of phyto abundance over 3 years (Fig. 5); broad taxonomic relative abundance distribution (dinos, diatoms, cyanos, chrysos) over all stations, pooled for the 3 years (Fig. 6), 2D PCA diagram of the stations (Fig. 7); 2D RDA diagrams (Fig. 8). These analyses are used by the authors to show clearly that the study area divides into 4 groups, A, B, C and D. In Fig 9 whisker boxes are used to show the distribution of T, S and VSI in each group, which is a very nice feature, and very clearly presented. Fig. 10 is used to present results of the SEM, indicating the statistically computed quantified effects (apparent effects?) of T and S on VSI (of course) and of T, S and VSI on DIN, DIP and phytoplankton. This is very original, as far as I am aware. Fig. 11 is used to explore the effects of DIN and DIP (particularly the N:P ratio, on phytoplankton of the four major taxa in the 4 regions at 3 depths (surface, DCM, and 200 m). It clearly shows different effects of N and P on the phytoplankton community structure in the different ecosystems corresponding to these three chosen depths.

There follows a Discussion rich, original and well argued.

Response: Dear Reviewer, thank you very much for your professional and careful review of our manuscript and for giving constructive comments.

However, in contrast to all this quality the introduction is terrible, and totally inappropriately targeted. While the authors have done an excellent job for the Methods, the Results and the Discussion, they need to scrap the Introduction completely and write it again. The same goes for the Abstract. If this is done well, this manuscript would constitute an important and original contribution.

Response: Dear Reviewer, thank you very much for your professional and careful review of our manuscript and for giving constructive comments. As your suggestion, we have reworked the abstract and introduction sections. Please refer to our revised

manuscript.

**Specific aspects**

Title

OK

Response: Dear Reviewer, thank you very much for your approval and encouragement.

Abstract

TERRIBLE.

Response: Dear Reviewer, we are very sorry for our poor description. As your suggestion, we have carefully reworked the abstract section of our manuscript in the revision process.

Introduction

TERRIBLE. (See above).

Response: Dear Reviewer, we are very sorry for our poor description. As your suggestion, we have carefully reworked the introduction section of our manuscript in the revision process.

**Materials and Methods**

THIS SECTION IS MOSTLY EXCELLENT.

Response: Dear Reviewer, thank you very much for your kind approval and encouragement.

P3L26 "PE" > "polyethylene"

Response: Dear Reviewer, thank you very much for your valuable and constructive comments. We have revised "PE" into "polyethylene".

P4L11 Insert reference for the Utermöhl method.

Response: Dear Reviewer, thank you very much for your professional and careful review of our manuscript and for giving constructive comments. We inserted the reference for the Utermöhl method.

L11-20 In this section state the minimum size of organisms identified and counted

Response: Dear Reviewer, thank you very much for your valuable and constructive

comments. The minimum size of the organisms identified and counted is 20 μm.

L23 "AA3 (SEAL., German(y)" Give bibliographic reference or web site.

Response: Dear Reviewer, thank you very much for your professional and careful review of our manuscript and for giving constructive comments. We added bibliographic references and rewrote the sentences. "The Technicon AA3 Auto-Analyzer (Bran + Luebbe, Norderstedt, Germany) based on classical colorimetric methods was used for the analysis and determination nutrient (Grasshoff et al., 2009)." Grasshoff, K., Kremling, K., Ehrhardt, M. (Eds.).: Methods of Seawater Analysis; John Wiley & Sons: Hoboken, NJ, USA; ISBN3-527-29589-5, 2009.

P5L15 "average" of temperature and salinity: give the precise dataset for which the average was computed.

Response: Dear Reviewer, I apologized for any inconvenience caused to your review, due to my clerical error. We are very sorry for our poor description, we have reworded the sentence "S and T are the salinity and temperature, respectively, and Sref and Tref are the temperature and salinity at 5 m, ΔT is equal to 0.5 °C."

L28 "the three years"

Response: Dear Reviewer, thank you very much for your valuable and constructive comments. We have revised "for three years" into "the three years".

**Results**

THIS SECTION IS MOSTLY EXCELLENT.

Response: Dear Reviewer, thank you very much for your kind approval and encouragement.

P7L9 "the same" > "a similar"

Response: Dear Reviewer, thank you very much for your valuable and constructive comments. We have revised "the same" into "a similar".

P8L5 Delete "variation in" (repetition)

Response: Dear Reviewer, thank you very much for your valuable and constructive comments. We have deleted "variation in".

L12 "variability" > "variation"

Response: Dear Reviewer, thank you very much for your valuable and constructive comments. We have revised "variability" into "variation".

L16 "showed a relatively uniform" > "varied little from year to year in their"

Response: Dear Reviewer, thank you very much for your valuable and constructive comments. We have revised "showed a relatively uniform" into "varied little from year to year in their".

L17-18 "extending..." > with a minor abundance peak at about 10°N."

Response: Dear Reviewer, thank you very much for your valuable and constructive comments. We have revised "extending in latitude, especially between the equator to 10 °N" into "with a minor abundance peak at about 10°N".

L18-19 "The abnormally..." > "This abundance peak was associated..."

Response: Dear Reviewer, thank you very much for your valuable and constructive comments. We have revised "The abnormally high phytoplankton abundance in this region is associated" into "This abundance peak was associated".

L20 "observed also..."

Response: Dear Reviewer, thank you very much for your valuable and constructive comments. We have revised "observed" into "observed also".

L21 "in southern Taiwan" > "south of Taiwan"

Response: Dear Reviewer, thank you very much for your valuable and constructive comments. We have revised "in southern Taiwan" into "south of Taiwan".

P9L1 Delete "As can be seen from the figure,"

Response: Dear Reviewer, thank you very much for your valuable and constructive comments. We deleted "As can be seen from the figure,".

L3 "regional variations in latitude" > "variations with latitude"

Response: Dear Reviewer, thank you very much for your valuable and constructive comments. We have revised "regional variations in latitude" into "variations with latitude".

L5-6 Delete ", and..." (Repetition)

Response: Dear Reviewer, thank you very much for your valuable and constructive comments. We deleted "and phytoplankton abundance gradually decreased with increasing latitude".

L11-13 "...the lower phytoplankton abundance was mostly dominated by..."

Response: Dear Reviewer, thank you very much for your valuable and constructive comments. We have revised "the phytoplankton abundance was lower than that in" into "the lower phytoplankton abundance was mostly dominated by".

L19 "... little interannual difference between species, ..."

Response: Dear Reviewer, thank you very much for your valuable and constructive comments. We have revised "little difference in interannual changes between species," into "little interannual difference between species,".

L26 "4.8" > "4.8%"; "1.4" > "1.4%"

Response: Dear Reviewer, we are very sorry for your confusion due to our poor description. We rephrased the sentence "The species ratio of diatoms to dinoflagellates in Group A (dias: dinos = 4.8) was higher than that in Group B (dias: dinos = 1.4)."

L29-30 Delete "The horizontal.." It's already marked on the figure 7.

Response: Dear Reviewer, thank you very much for your valuable and constructive comments. We deleted "The horizontal and vertical axes explain 51.87% and 21.41% of the phytoplankton community structure, respectively.".

P11 L14 "methanogens" This seems to be a mistake.

Response: Dear Reviewer, I apologized for any inconvenience caused to your review, we have revised "methanogens" into "dinoflagellates".

L26 Insert "here," before "the phytoplankton"

Response: Dear Reviewer, thank you very much for your valuable and constructive comments. We inserted "here," before "the phytoplankton".

P12L6 "3.4" > "3.6"

Response: Dear Reviewer, I apologized for any inconvenience caused to your review, we have revised "3.4" into "3.6".

L8 "... of the sample from 5 m above" Seems to be a spurious insertion. Delete.
Response: Dear Reviewer, thank you very much for your professional and careful review of our manuscript and for giving constructive comments. We are very sorry for your confusion due to our poor description, we deleted "... of the sample from 5 m above".

L12-13 "The strong spatial variability..." > "Fig. 9 shows clear variation in T-S.".
Response: Dear Reviewer, thank you very much for your valuable and constructive comments. We have revised "The strong spatial variability of T-S was evident from the characteristics of salinity and temperature." into "Fig. 9 shows clear variation in T-S.".

L14-15 "number of VSIs" > "values of VSI"
Response: Dear Reviewer, thank you very much for your valuable and constructive comments. We have revised "number of VSIs" into "values of VSI".

L15 "was > "were"
Response: Dear Reviewer, thank you very much for your valuable and constructive comments. We have revised "was" into "were".

L16 Delete "There were... groups"
Response: Dear Reviewer, thank you very much for your valuable and constructive comments. We deleted "There were obvious differences between the four groups; that is,".

L18 "linearly fitted to temperature" > "related to temperature"
Response: Dear Reviewer, thank you very much for your valuable and constructive comments. We have revised "linearly fitted to temperature" into "related to temperature".

L19 Delete "The fitting results showed that the"
Response: Dear Reviewer, thank you very much for your valuable and constructive comments. We deleted "The fitting results show that the".

L20-21 Delete "It can be noted that the"

Response: Dear Reviewer, thank you very much for your valuable and constructive comments. We deleted "It can be noted that the".

L21 "more" > "most"

Response: Dear Reviewer, thank you very much for your valuable and constructive comments. We have revised "more" into "most".

P13L7 "3.6" > "3.7"

Response: Dear Reviewer, I apologized for any inconvenience caused to your review, we have revised "3.6" into "3.7".

L7 "parameter" > "parameters"

Response: Dear Reviewer, thank you very much for your valuable and constructive comments. We have revised "parameter" into "parameters".

P14L14 At the end of this light you may like to add, "and growth may have become increasingly limited by light."

Response: Dear Reviewer, thank you very much for your professional and careful review of our manuscript and for giving constructive comments. As your suggestion, we added "and growth may have become increasingly limited by light." to this section. Thank you again for your valuable comments!

**Discussion**

VERY GOOD DISCUSSION.

Response: Dear Reviewer, thank you very much for your kind approval and encouragement.

P15L1 "Kuroshio" > "The Kuroshio"

Response: Dear Reviewer, thank you very much for your valuable and constructive comments. We have revised "Kuroshio" into "The Kuroshio".

L1 "WPWP" Add this to Fig. 1.

Response: Dear Reviewer, thank you very much for your valuable and constructive

comments. We added "WPWP" to the Fig. 1.

L1 After "interaction" insert "and climate modulation"
Response: Dear Reviewer, thank you very much for your valuable and constructive comments. We inserted "and climate modulation" after "interaction".

L14 "the vertical trawl" > "vertical hauls"
Response: Dear Reviewer, thank you very much for your valuable and constructive comments. We have revised "the vertical trawl" into "vertical hauls".

L24 "acquisition of nutrient strategies" > "nutrient acquisition strategies"
Response: Dear Reviewer, thank you very much for your valuable and constructive comments. We have revised "acquisition of nutrient strategies" into "nutrient acquisition strategies".

L25-26 "... dinoflagellates use mixotrophy, engulfing prey as well as feeding using peduncles and palia, while phosphorus..."
Response: Dear Reviewer, thank you very much for your valuable and constructive comments. We have revised "…dinoflagellates have the ability of mixotrophy, and the mixotrophic modes of dinoflagellates include direct engulfment of prey, peduncle feeding, and pallium feeding, and phosphorus…" into "…dinoflagellates use mixotrophy, engulfing prey as well as feeding using peduncles and palia, while phosphorus…".

L28 "was > "is"
Response: Dear Reviewer, thank you very much for your valuable and constructive comments. We have revised "was" into "is".

P16L2 "Fig. 11" > "Fig. 6"
Response: Dear Reviewer, thank you very much for your valuable and constructive comments. We have revised "Fig. 11" into "Fig. 6".

L3-4 "... has already been demonstrated (Grosskopf et al., 2012; ..."
Response: Dear Reviewer, thank you very much for your valuable and constructive comments. We have revised "…has been demonstrated several years ago" into "... has

already been demonstrated".

L4 "The presence of slight" > "The virtual absence"

Response: Dear Reviewer, thank you very much for your valuable and constructive comments. We have revised "The presence of slight" into "The virtual absence".

L9 "were" > "are"

Response: Dear Reviewer, thank you very much for your valuable and constructive comments. We have revised "were" into "are".

L9 "susceptible" > "affected"

Response: Dear Reviewer, thank you very much for your valuable and constructive comments. We have revised "susceptible" into "affected".

L11 "across" > "along"

Response: Dear Reviewer, thank you very much for your valuable and constructive comments. We have revised "across" into "along".

L12 "indicated" > "indicates"

Response: Dear Reviewer, thank you very much for your valuable and constructive comments. We have revised "indicated" into "indicates".

L13 "indicated" > "indicates"

Response: Dear Reviewer, thank you very much for your valuable and constructive comments. We have revised "indicated" into "indicates".

L16 "competition is reduced as light limitation kicks in, and the nutrient ratio approaches...

Response: Dear Reviewer, thank you very much for your valuable and constructive comments. We have revised "competition reduced and the nutrient ratio approached" into "competition is reduced as light limitation kicks in, and the nutrient ratio approaches".

L17 "nutrients partly affected..." > "nutrient ratios thus affect..."

Response: Dear Reviewer, thank you very much for your valuable and constructive

comments. We have revised "nutrients partly affected" into "nutrient ratios thus affect".

L20 Around here, it might be good to very briefly mention the possibility of limitation by other nutrients such as iron. Also mention, if you like, that some of the phytoplankton sampled may have recently sunk from upper layers, and therefore represent the nutrient rations and T-S of these layers. You, the authors, may have a feeling for this in the present work.

Response: Dear Reviewer, thank you very much for your professional and careful review of our manuscript and for giving constructive comments. We have added relevant content in this section "Iron is essential for the synthesis of nitrogen-fixing enzymes in *Trichodesmium*, and *Trichodesmium* have a higher demand for iron than other planktonic organisms. The main source of iron in open ocean is atmospheric deposition. Duce et al. showed that the flux of iron deposition is higher in the WPO, so iron is an important environmental limiting factor for the growth of *Trichodesmium* after temperature (Duce and Tindale, 1991). And we suggest that some of the sampled phytoplankton may have recently sunk from the upper layers and therefore represent nutrient rationing and T-S in the water layers. Directly sinking phytoplankton cells are major contributors to surface carbon export and an important component of ocean carbon sink (Boyd and Newton, 1999). The phytoplankton cells can regulate their sinking rates in a variety of ways, such as the physiological state of themselves (Eppley et al., 1967), morphology of themselves (Pitcher et al., 1989), light (Bienfang, 1985) and environmental factors such as temperature and nutrients (Titman and Kilham, 1976).

L22-24 Delete "With global... structure". It's too speculative.
Response: Dear Reviewer, thank you very much for your valuable and constructive comments. We have deleted this sentence.

L24 Delete "typical"
Response: Dear Reviewer, thank you very much for your valuable and constructive comments. We have deleted "typical".

L24 "severe" > "strong"
Response: Dear Reviewer, thank you very much for your valuable and constructive comments. We have revised "severe" into "strong".

L25-28 "... the interannual variation of phytoplankton was not significant. It remained stably oligotrophic, and the vertical stratification structure determined that of environmental resources such as nutrients, thus forming four contrasting environments, each with its characteristic phytoplankton community structure." [I think you can't say that stratification produced the T-S environment. In any case there is no need to say it.]

Response: Dear Reviewer, thank you very much for your professional and careful review of our manuscript and for giving constructive comments. We apologize for the inconvenience caused to your review and we have reworked the sentence according to your suggestion. Thanks again for your professional and careful suggestions.

L31-34 "... from the deep layer below the thermocline, which affects the N:P ratio, and restricts vertical migration as well as physiologically affecting the vertical structure of phytoplankton growth and mortality."

Response: Dear Reviewer, we are very sorry for our poor description. As your suggestion, we have carefully reworked the sentence. Again, we deeply appreciate your efficient and professional review of our manuscript.

L37 "has been" > "is"

Response: Dear Reviewer, thank you very much for your valuable and constructive comments. We have revised "has been" into "is".

L38 Delete "since the 1960s"

Response: Dear Reviewer, thank you very much for your valuable and constructive comments. We have deleted "since the 1960s".

L39 "is suitable for living" > "thrives"

Response: Dear Reviewer, thank you very much for your valuable and constructive comments. We have revised "is suitable for living" into "thrives".

P17L1 "Trichodesmium" needs italics

Response: Dear Reviewer, thank you very much for your valuable and constructive comments. We italicized "*Trichodesmium*".

L2 "believe" > "have proposed"

Response: Dear Reviewer, thank you very much for your valuable and constructive

comments. We have revised "believe" into "have proposed".

L4 "believe" > "suggest"
Response: Dear Reviewer, thank you very much for your valuable and constructive comments. We have revised "believe" into "suggest".

L5 "also is consistent"
Response: Dear Reviewer, thank you very much for your valuable and constructive comments. We have revised "is consistent" into "also is consistent".

L6 "where the temperature was not restricted," > "where the surface temperatures all exceeded 20°C,"
Response: Dear Reviewer, thank you very much for your valuable and constructive comments. We have revised "where the temperature was not restricted," into "where the surface temperatures all exceeded 20°C,".

L7 "higher than in those with lower temperatures"
Response: Dear Reviewer, thank you very much for your valuable and constructive comments. We have revised "higher than that at relatively low temperatures" into "higher than in those with lower temperatures".

L8-21 based on the data you present and other knowledge, the present referee is not entirely convinced by the authors' arguments, but the authors should have the right to interpret their data in this way if they so choose.
Response: Dear Reviewer, we are very sorry for our poor description. As your suggestion, we have delated the inappropriate description and reworked the sentence "Temperature not only directly affected phytoplankton growth, but also indirectly affected phytoplankton growth and abundance by regulating VSI to drive the nutrient ratio (N:P) (Figure 10).". Again, we deeply appreciate your efficient and professional review of our manuscript.

L10 "high-value area" > "high abundance"
Response: Dear Reviewer, thank you very much for your valuable and constructive comments. According to your suggestion, we have redescribed this section so that "high-value area" has been deleted.

L20 "severe" > "strong"

Response: Dear Reviewer, thank you very much for your valuable and constructive comments. According to your suggestion, we have redescribed this section so that "severe" has been deleted.

L25-27 "A strong coupling exists among the nutrient supply rate, the photosynthetic performance of phytoplankton (Bouan et al., 2006), the phytoplankton biomass and primary production, particularly in eutrophic areas (Richardson et al., 2019)." Delete "which directly limits nutrient supply"

Response: Dear Reviewer, thank you very much for your insightful and constructive comments. As your suggestion, we have reworked the sentence and deleted "which directly limits nutrient supply". Thank you again for your valuable comments!

L28 "causes" > "effects"

Response: Dear Reviewer, thank you very much for your valuable and constructive comments. We have revised "causes" into "effects".

L28 "in" > "on"

Response: Dear Reviewer, thank you very much for your valuable and constructive comments. We have revised "in" into "on".

L34 "which" > "and"

Response: Dear Reviewer, thank you very much for your valuable and constructive comments. We have revised "which" into "and".

L35 "demonstrated" > "demonstrate"

Response: Dear Reviewer, thank you very much for your valuable and constructive comments. We have revised "demonstrated" into "demonstrate".

L37 Delete "the survival of"

Response: Dear Reviewer, thank you very much for your valuable and constructive comments. We have deleted "the survival of".

L38 "poor activity" Do you mean "low mobility"?

Response: Dear Reviewer, thank you very much for your valuable and constructive comments. We have revised "poor activity" into "low mobility".

L38-39 "in the circulation and water with high nutrient content" > "in mixed water with high nutrient content"

Response: Dear Reviewer, thank you very much for your valuable and constructive comments. We have revised "poor activity" into "low mobility".

L40 After "C and D regions", insert "(Fig. 9b)"

Response: Dear Reviewer, thank you very much for your valuable and constructive comments. We have inserted "(Fig. 9b)" after "C and D regions".

L40 "the" > "relative"

Response: Dear Reviewer, thank you very much for your valuable and constructive comments. We have revised "the" into "relative".

L41-43 I don't understand what you mean in this sentence.

Response: Dear Reviewer, we are very sorry for your confusion due to our poor description. We wanted to express that dinoflagellate have more abundance in Groups C and D with weaker vertical stratification. We deleted this sentence due to our inaccurate description.

P18 L4 Delete "which was not discussed in this article"

Response: Dear Reviewer, thank you very much for your valuable and constructive comments. We have deleted "which was not discussed in this article".

L5 "further investigated"

Response: Dear Reviewer, thank you very much for your valuable and constructive comments. We have revised "future studies" into "further investigated".

**Conclusions**

P18L9 Delete "typical"

Response: Dear Reviewer, thank you very much for your valuable and constructive comments. We have deleted "typical".

L14 "variability" > "variation"

Response: Dear Reviewer, thank you very much for your valuable and constructive comments. We have revised "variability" into "variation".

L15 "Structural Equation Model (SEM)"

Response: Dear Reviewer, thank you very much for your valuable and constructive comments. We have revised "SEM" into "Structural Equation Model (SEM)".

L17 After "oligotrophic", insert "sea area"

Response: Dear Reviewer, thank you very much for your valuable and constructive comments. We have inserted "sea area" after "oligotrophic".

**Acknowledgements**

L26 "We thank"

Response: Dear Reviewer, thank you very much for your valuable and constructive comments. We have revised "Thank" into "We thank".

**References**

I not that given names and family names are inverted in the first reference ,"Mitra, A. and Flynn, K.J.". Please check all references.

Response: Dear Reviewer, we are very sorry for our poor description. As your suggestion, we have carefully checked all references.

**Figures, Tables**

The figures and tables are all very good.

Response: Dear Reviewer, thank you very much for your kind approval and encouragement.

Fig. 7 needs more contrast.

Response: Dear Reviewer, we would like to express our great appreciation to your time and effort in reviewing our manuscript, and your insightful and constructive comments helped to improve the accuracy and quality of our manuscript significantly. We adjusted the color scheme in Figure 7 in hopes of increasing the contrast.

Special thanks to you for your good comments!

Dear Editors and Reviewers:

Thank you for your letter and for the reviewers' comments concerning our manuscript entitled "Vertical stratification-driven nutrient ratios regulate phytoplankton community structure in the oligotrophic western Pacific Ocean" (ID: os-2021-67). Those comments are all valuable and very helpful for revising and improving our paper, as well as the important guiding significance to our researches. We have studied comments carefully and have made correction which we hope meet with approval. Revised portion are marked in red in the paper. We have a small doubt that maybe Reviewer #2 saw our first-versions manuscript instead of the revised-versions manuscript in this review process. This must be our cause that our revised-versions manuscript is not conspicuous enough, and we hope that we will have the opportunity to present the revised draft to Reviewer #2 in this time. If there is anything else we should do, please don't hesitate to let us know. Again, we deeply appreciate your efficient and professional review of our manuscript. The main corrections in the paper and the responds to the reviewer's comments are as flowing:

Responds to the reviewer's comments:
Reviewer #2:

Page 1 Line 40 heat chain is a new word for me. Please check whether there is this word in oceanography.
Response: Dear Reviewer, thank you very much for your kind letter and encouragement. We also appreciate the time and effort that you have dedicated to providing valuable comments and suggestions, which helped us to improve the quality of our manuscript significantly. We have carefully studied these insightful comments and have made corrections which we hope meet with your approval. We are very sorry for our poor description. We have a small doubt that maybe Reviewer #2 saw our first-versions manuscript instead of the revised-versions manuscript in this review process. This must be our cause that our revised-versions manuscript is not conspicuous enough, and we hope that we will have the opportunity to present the revised manuscript to Reviewer #2 in this time. These issues appeared in the first-versions manuscript and have been revised in our revised-versions manuscript. As your suggestions, we have carefully reworked the introduction section of our manuscript in the revision process. We have deleted these incorrect statements.

Page 2 line 10-12, this sentence is not complete. Line 15 wind-induced, here strange English. Line 19 thermal slope, please check.

Response: Dear Reviewer, I apologized for any inconvenience caused to your review. We are very sorry for our poor description. As your suggestion, we have carefully reworked the introduction section of our manuscript in the revision process. We have deleted these incorrect statements.

Page 3 line 7, Kexue. Sampling dates of the three cruises were preferred. Line 10-12, unreadable. Line 25, unreadable.

Response: Dear Reviewer, thank you very much for your professional and careful review of our manuscript and for giving constructive comments. As your suggestion, we have reworked this section "This study relied on the shared voyage of the WPO (0– 20 °N, 120–130 °E), commissioned by the National Natural Science Foundation of China. Physical, biological, chemical, and geological surveys were carried out from September to November in 2016, 2017, and 2018 aboard the R/V *Kexue*. The sampling stations used in this study are shown in Figure 1; the sampling layers were 5, 25, 50, 75, 100, 150, and 200 m. Phytoplankton samples from different water layers were placed in 1 L polyethylene bottles, fixed in formaldehyde solution (3%), and stored in dark. Nutrient samples from different layers were placed in PE bottles, frozen, and stored at −20 °C for laboratory nutrient analysis."

Figure 1, the small figures a b c d were not explained. STCC, NEC etc were not explained. I did not find red, yellow and green triangles and black dots in this figure.

Response: Dear Reviewer, thank you very much for your valuable and constructive comments. As your suggestion, we have reworked the description of Figure 1 "Figure 1. Stations in the western Pacific Ocean (WPO) of three cruises. (a): Current systems of the WPO; (b), (c), and (d): sampling stations of 2016, 2017 and 2018 cruises, respectively. The station at 130°E forms the section A, and the station at 20°N forms the section B. Map of the WPO shows the major geographic names and the surface currents, including the Subtropical Counter Current (STCC), the North Equatorial Current (NEC), the Northern Equatorial Counter Current (NECC), the South Equatorial Current (SEC), the New Guinea Coastal Current (NGCC), the Mindanao Current (MC), the Mindanao Eddy (ME), the Halmahera Eddy (HE).".

Figure 5. a b c d were not explained. What is the dots, lines in each small figure?

Response: Dear Reviewer, thank you very much for your valuable and constructive comments. We have deleted the original Figure 5 and replaced it with Figures 4 and 5 in the new manuscript to better show the horizontal and vertical distribution of phytoplankton abundance in space.

Table 3, Date should be month.

Response: Dear Reviewer, thank you very much for your valuable and constructive comments. We have revised "Date" into "Month" in the Table 3.

References. All the references listed were not aligned by alphabetic order. Strange.

Response: Dear Reviewer, we are very sorry for our poor description. As your suggestion, we have carefully checked all references.

Special thanks to you for your good comments!